# Dynamics Are Learned, Not Told: Semi-Supervised Discovery of Latent Dynamics Geometries For Zero-Shot Policy Adaptation

Zhiming Xu[1]   Weitao Zhou[2 3]   Xianghui Pan[4]   Nanshan Deng[2 5]   Chengju Liu[4]   Qijun Chen[4]   Chenpeng Yao[4]

## Abstract

Real-world dynamics shifts pose a critical challenge for reinforcement learning in robotics, as policies tightly coupled to nominal environments often fail catastrophically when physical conditions change. Most existing methods rely on encoding explicitly identified physical parameters into a latent context, a parameter-centric paradigm that depends on pre-specified axes of variation and becomes brittle under unmodeled or compound dynamics changes. We revisit dynamics adaptation from an outcome-centric perspective: rather than telling policies what the dynamics are, we enable them to learn how dynamics affect interaction outcomes. Theoretically, this is grounded in a monotonic relationship between target-domain regret and the Lipschitz constant of a trajectory dynamics encoder. Practically, this constant can be upper-bounded through contrastive learning, yielding a smooth, task-relevant latent topology without privileged dynamics information. On MuJoCo benchmarks, our method consistently outperforms parameter-centric baselines under severe dynamics shifts, including unmodeled and time-varying parameters, while also improving in-distribution stability and latent interpretability. Overall, these results validate that controlling latent geometry is a principled mechanism for robust adaptation.

## 1. Introduction

Model-free Deep Reinforcement Learning (DRL) excels at mastering complex control tasks by exploiting the specific

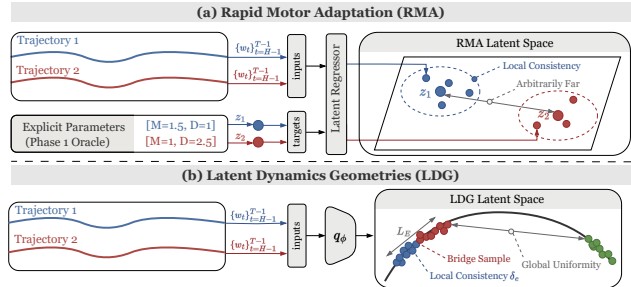

*Figure 1.* Conceptual comparison of adaptation paradigms. (a) RMA (parameter-centric) uses a trajectory encoder to regress oracle output ($z_1$ and $z_2$), which are functionally similar trajectories but are mapped to arbitrary distances since their parameters differ. (b) Our method LDG (outcome-centric) learns a latent dynamics geometry directly from trajectory outcomes. By enforcing local consistency ($\delta_e$) and global uniformity via contrastive learning, we construct a smooth manifold (characterized by bounded small $L_E$) that enables robust zero-shot adaptation.

transition dynamics of the training environment. Through trial and error, agents internalize precise correlations between actions and state evolution, for instance, exactly how much momentum is required to brake safely. However, this specialization becomes a liability when physical properties shift, such as when a robot carries an unknown payload or suffers mechanical wear. In these Out-Of-Distribution (OOD) scenarios, policies optimized for the nominal dynamics often fail catastrophically; a braking maneuver valid for a light robot may result in collision for a heavier one.

A prominent line of work, represented by Rapid Motor Adaptation (RMA (Kumar et al., 2021)), addresses this challenge by conditioning policies on a compression of system dynamics parameters. During deployment, this latent context is inferred from short interaction histories by an adaptation module (Kumar et al., 2021; 2022; Qi et al., 2023; Liang et al., 2024). While effective in practice, this paradigm is inherently *parameter-centric*: rather than allowing the policy to learn to select the most relevant features end-to-end, this approach imposes a manual inductive bias that effectively hard-codes the feature space to pre-specified axes of variation.

We argue this parameter-centric view obscures two fundamental intricacies. First, real robotic systems are subject

---
[1]School of Computer Science and Engineering, Tongji University, Shanghai, China [2]School of Vehicle and Mobility, Tsinghua University, Beijing, China [3]Simple AI, Beijing, China [4]Department of Electronics and Information Engineering, Tongji University, Shanghai, China [5]Rimbot, Beijing, China. Correspondence to: Weitao Zhou, Chenpeng Yao <zhouwt801@gmail.com, yaochenpeng@tongji.edu.cn>.

*Proceedings of the 43rd International Conference on Machine Learning*, Seoul, South Korea. PMLR 306, 2026. Copyright 2026 by the author(s).

to a wide range of unmodeled, coupled (Eysenbach et al., 2021; Guo et al., 2024), and time-varying factors (Zhu et al., 2025), for which no fixed set of physical parameters provides a complete description. Moreover, distinct physical phenomena often induce indistinguishable effects on motion; for example, increased payload and higher friction all manifest similarly as "resistance to acceleration". Consequently, the quantities most relevant to adaptation are not the physical parameters themselves, but their realized effects on interaction outcomes.

Motivated by this observation, we adopt an *outcome-centric* approach to adaptation. By integrating contrastive learning (Chen et al., 2020) into a variational inference framework (Higgins et al., 2017a), we explicitly enforce invariance within the same interaction regime while preserving separability across regimes that demand different control responses. We further investigate how latent space perturbations translate to output instability, and show theoretically that controlling representation sensitivity, characterized by the encoder's Lipschitz constant, upper-bounds the sub-optimality of cross-domain adaptation. Crucially, naively enforcing smoothness is insufficient; successful adaptation requires simultaneous enforcement of smoothness and topological structure within the latent space. We show that contrastive learning naturally promotes such a topology, characterized by local consistency and global uniformity (Fig. 1(b)). From a practical perspective, contrastive learning effectively filters nuisance variables via gradient orthogonality, ensuring the latent space encodes only task-relevant dynamics, a desired property for latent-conditioned policies (Zhang et al., 2021; Liu et al., 2023) but often absent in purely reconstructive baselines. Empirically, this geometry-aware representation improves both In-Distribution (ID) stability and zero-shot generalization across diverse OOD environments, including settings with unmodeled and time-varying physical variations.

## 2. Related Work

**Dynamics Adaptation.** Adaptation to dynamics shifts is a central challenge in robotics, where transition dynamics in the target environment differ from the training domain. Domain Randomization (DR (Tobin et al., 2017; Peng et al., 2018)) trains a single policy across diverse dynamics but may sacrifice optimality. Off-Dynamics Reinforcement Learning (Eysenbach et al., 2021) utilizes limited target domain interactions to amend the source policy, often via domain classifiers that discourage reliance on source-specific dynamics (Eysenbach et al., 2021; Guo et al., 2024). Meta-RL approaches learn dynamics priors for rapid adaptation (Nagabandi et al., 2019), but require online gradient updates at deployment. Beyond robustness, another line of work seeks stronger zero-shot target per-

formance under physical-parameter shifts by adapting the policy using a compact context that captures the underlying dynamics variation. A representative explicit paradigm, exemplified by Rapid Motor Adaptation (RMA (Kumar et al., 2021)), uses supervision over dynamics parameters (e.g., mass and friction) and maps them to latent variables to modulate the policy. Generation of these latent variables during deployment are often achieved via an adaptation module that estimates latents from trajectory histories (Kumar et al., 2021; 2022; Qi et al., 2023; Liang et al., 2024) or test-time optimization (Yu et al., 2019; Hu et al., 2025). This parameter-centric design can become brittle when dynamics shifts are unmodeled (Eysenbach et al., 2021; Guo et al., 2024) or time-varying (Zhu et al., 2025), moreover this reliance on privileged information proves complicated or even infeasible when external objects exists (Qi et al., 2023; Liang et al., 2024; He et al., 2023). In contrast, implicit approaches infer dynamics-relevant context directly from interaction histories, for example, via trajectory dynamics encoders (Lee et al., 2020b) or latent optimization (Yu et al., 2020). Building upon implicit approaches, we further investigate how latent space perturbations translate to output instability, drawing parallels to stability challenges in image generation (Yoshida & Miyato, 2017; Miyato et al., 2018; Brock et al., 2018; Song et al., 2023). Extending this framework to policy learning, we reveal that controlling network smoothness is a fundamental principle shared across generative vision and policy adaptation. Crucially, our method simultaneously enforces smoothness and latent geometry for robust adaptation.

**Latent Inference For Control.** Many implicit adaptation methods rely on variational inference (Kingma & Welling, 2014) to construct a compact context for decision making, following the success of latent dynamics models in image-based domains (Watter et al., 2015; Hafner et al., 2019). While our work also learns latent dynamics models, the latent variables in our setting represent dynamics variation rather than low-dimensional image compression. In this sense, our formulation is closer to skill-based latent representations (Eysenbach et al., 2019; Sharma et al., 2020), where each latent conditions a distinct expert behavior, and recent works have further extended these ideas toward dynamics-aware formulations (Liu et al., 2021; 2025). Related ideas also appear in interaction settings, where latent variables encode beliefs over intent from histories (He et al., 2023; Xie et al., 2021; Parekh et al., 2022). Similarly, Meta-RL uses inference networks to summarize interaction histories into latent belief states for faster adaptation (Finn et al., 2017; Rakelly et al., 2019; Zintgraf et al., 2020). Inspired by these trajectory-based inference frameworks, we adopt a probabilistic latent approach for modeling dynamics variation, which exhibits longer temporal dependencies and substantially higher intrinsic dimensionality than intent

or task context. This highlights the need to go beyond inference alone for dynamics adaptation.

**Geometry-Regularized Representations.** Representation smoothness and consistency is a well-investigated topic in the field of image generation to enhance training stability and sample quality (Yoshida & Miyato, 2017; Miyato et al., 2018; Brock et al., 2018; Song et al., 2023). In the context of dynamics adaptation, latent sensitivity governs how trajectory-level distribution shifts map to latent perturbations and thus closed-loop policy deviation, making objectives that regularize neighborhood structure and smoothness crucial for robust adaptation (Higgins et al., 2017b; Van der Pol et al., 2020; Zhang et al., 2021). While Variational AutoEncoders (VAEs (Kingma & Welling, 2014)) provide a tractable framework, the standard variational lower bound primarily optimizes for reconstruction fidelity and may overlook global latent topology (Wang & Isola, 2020). This limitation has motivated a shift toward objectives that more directly shape latent structure. CURL (Laskin et al., 2020) demonstrates that contrastive instance discrimination yields more sample-efficient control representations than generative modeling, and SPR (Schwarzer et al., 2021) shows that enforcing multi-step temporal consistency in the latent space outperforms the standard variational objective for policy learning. However, these methods mainly study representation quality and sample efficiency, and to the best of our knowledge do not explicitly connect latent geometry to zero-shot dynamics adaptation performance. Drawing inspiration from prior works that translates bisimulation metrics in state space to distances in latent space to increase policy robustness (Zhang et al., 2021; Liu et al., 2023), our work captures physical semantics to shape latent space geometry, enabling zero-shot dynamics adaptation.

## 3. Preliminaries

Throughout this paper, we use subscript $T$ and $S$ to denote quantities in the target domain and source domain respectively, such that $V_S^{\pi_S}$ denotes the value function for source policy $\pi_S$ in the source domain. We consider two Markov Decision Processes (MDPs), $\mathcal{M}_S$ and $\mathcal{M}_T$, that share the same state space, action space and reward function but differ only in transition dynamics $p_S$ and $p_T$, induced by changes in physical parameters (for a more rigorous definition, refer to Appendix A.1.1). Given a trajectory $\tau = \{s_i, a_i\}_{i=1}^{L-1}$, define the length-$H$ window $w_t = \{s_i, a_i\}_{i=t-H}^{t-1}$, from which task-relevant representations are inferred via an encoder, given by $z_t = E(w_t)$. We learn in $\mathcal{M}_S$ an encoder $E$ and latent-conditioned policy $\pi(\cdot|s, z_S)$ that generalizes to $\mathcal{M}_T$ by utilizing the corresponding latent dynamics context $z_T$.

Let $\rho_t(\pi, p)(w)$ be the marginal distribution of $w_t$ and

$d_t^{\pi,p}(s_t)$ be the probability of being in state $s_t$ at time step $t$ under policy $\pi$ and dynamics $p$ (see Appendix A.1.2 for their analytical forms). The following definitions are established to aid subsequent analysis.

**Definition 1.** Two MDPs $\mathcal{M}_S$ and $\mathcal{M}_T$ are $\epsilon_p$-close in dynamics, if they share the same state space, action space and reward function, but differ in transition dynamics $p_S(s'|s, a)$ and $p_T(s'|s, a)$, such that:

$$D_{TV}(p_T(\cdot|s, a), p_S(\cdot|s, a)) \leq \epsilon_p \quad \forall(s, a) \quad (1)$$

**Definition 2.** A latent-conditioned policy $\pi(\cdot|s, z)$ is $L_\pi$-smooth, if there exists a constant $L_\pi$ such that:

$$\sup_s D_{TV}(\pi(\cdot|s, z_S), \pi(\cdot|s, z_T)) \leq L_\pi \|z_S - z_T\|_2 \quad (2)$$

**Definition 3.** For a fixed policy $\pi$ and transition dynamics $p$, the time-indexed latent centroid $\mu_t(\pi, p)$ is the expected encoding of the window at that specific time step:

$$\mu_t(\pi, p) = \mathbb{E}_{w \sim \rho_t(\pi, p)}[E(w)] \quad (3)$$

**Definition 4.** An encoder is $\delta_e$-consistent if:

$$\sup_t \sup_{w \sim \rho_t(\pi, p)} \|E(w) - \mu_t(\pi, p)\|_2 \leq \delta_e \quad (4)$$

**Definition 5.** The encoder Lipschitz constant $L_E$ with respect to the total variation divergence of the induced window distributions is defined as:

$$L_E = \sup_t \sup_{(\pi, p), (\tilde{\pi}, \tilde{p})} \frac{\|\mu_t(\pi, p) - \mu_t(\tilde{\pi}, \tilde{p})\|_2}{D_{TV}(\rho_t(\pi, p), \rho_t(\tilde{\pi}, \tilde{p}))} \quad (5)$$

The encoder is considered $L_E$-smooth if $L_E < \infty$.

**Definition 6.** Let $\mathcal{W}$ be the space of trajectory windows, the support of physically feasible windows under dynamics $p$ and policy $\pi$ is defined as:

$$\text{supp}(\pi, p) = \{w \in \mathcal{W} \mid \forall(s, a, s') \in w, p(s'|s, a) > 0\} \quad (6)$$

## 4. Methods

### 4.1. Performance Gap Under Dynamics Shift

We relate the performance of an adaptive policy to that of an oracle policy optimized in the target MDP, and present the following theorem to reveal the relationship between sub-optimality gap (regret) and encoder properties.

**Assumption 1.** There exists a constant $C_d < 1$ such that $\sup_t D_{TV}(d_t^{\pi_S, p_S}(s_t), d_t^{\pi_T, p_T}(s_t)) \leq C_d$.

*Remark* 1. This assumption is theoretically justified by the ergodicity of the underlying MDP, where the transition operator induced by a fixed policy and dynamics acts as a contraction mapping on the space of probability measures.

**Assumption 2.** The latent-conditioned policy $\pi_S = \pi(\cdot|s, z_S)$ approximates the source oracle $\pi_S^* = \pi(\cdot|s)$ within a bounded error $\delta_{train}$ during training.

**Theorem 1** (Target Domain Regret Bound). *Let $\mathcal{M}_S$ and $\mathcal{M}_T$ be two MDPs that are $\epsilon_p$-close in dynamics. If policy $\pi$ is $L_\pi$-smooth, encoder $E$ is $L_E$-smooth, $\delta_e$-consistent in both MDPs, Assumption 1 and 2 hold, then the bound for target domain regret is monotonically increasing with respect to $\delta_e$ and $L_E$.*

Proof is provided in Appendix A.5, within which we also derive a closed-form bound for target regret (Eq. (A.47)). We utilized the following decomposition during the proof:

$$
\left\| V_T^{\pi_T^*} - V_T^{\pi_T} \right\|_\infty \leq \underbrace{\left\| V_T^{\pi_T^*} - V_S^{\pi_S^*} \right\|_\infty}_{\text{(I) Oracle Difference}}
$$
$$
+ \underbrace{\left\| V_S^{\pi_S^*} - V_S^{\pi_S} \right\|_\infty}_{\text{(II) Training Error}} + \underbrace{\left\| V_S^{\pi_S} - V_T^{\pi_T} \right\|_\infty}_{\text{(III) Policy Adaptation}} \quad (7)
$$

which shows target regret is reduced when policy adaptation error (term (III)) is minimized. Furthermore, we prove that term (III) is monotonic with respect to encoder Lipschitz constant $L_E$, which controls how trajectory-level distributional shifts translate into latent perturbations. Intuitively, a smooth encoder (small $L_E$) maps trajectories generated by similar dynamics to nearby regions in the representation space, ensuring that small dynamics variations induce bounded changes in the latent code. This perceptual stability in turn promotes policy stability when the policy itself is smooth (Definition 2). For conventional VAE (Kingma & Welling, 2014), decoder reconstruction loss only force latent clusters under similar dynamics to be *different*, rather than *proximate*, in the latent space, essentially causing a larger $L_E$ and thus worse in-distribution stability. The ideology of translating dynamics similarity into latent vector distance is consistent with contrastive learning, where representations are formed by clustering similar (positive) samples and separating dissimilar (negative) samples. We elaborate this idea in the following subsection.

### 4.2. Contrastive Learning For Latent Geometry

In this subsection we first prove that optimizing the InfoNCE Loss (Chen et al., 2020) upper-bounds $L_E$, therefore tightens the bound for target regret according to Theorem 1, then explain how this loss helps distilling dynamics-related information from trajectory histories. Firstly, the InfoNCE Loss for a positive pair of examples $(i, j)$ is defined as:

$$
\ell_{i,j} = -\log \frac{\exp(\text{sim}(z_i, z_j)/\lambda)}{\sum_{k \neq i} \exp(\text{sim}(z_i, z_k)/\lambda)} \quad (8)
$$

where $\lambda$ is the temperature coefficient, and $\text{sim}(\cdot, \cdot)$ denotes a similarity measure, chosen as cosine similarity in this paper. Following the analysis by (Wang & Isola, 2020), the InfoNCE Loss asymptotically decomposes (as the number of negative samples $N \to \infty$) into Alignment and Uniformity:

$$
\lim_{N \to \infty} \mathcal{L}_{InfoNCE} \propto \mathbb{E}_{(w,w^-) \sim \rho_{data}} \left[ e^{-\|E(w)-E(w^-)\|_2^2} \right] +
$$
$$
\mathbb{E}_{(w,w^+) \sim \rho_{pos}} \left[ \|E(w) - E(w^+)\|_2^2 \right]
$$
$$
(9)
$$

where the first term is the Uniformity Loss $\mathcal{L}_{uniform}$, which encourages features to be uniformly distributed on a unit hypersphere, and the second term is the Alignment Loss $\mathcal{L}_{align}$ that encourages positive pair to be mapped to nearby features. Given this decomposition, we provide the following theorem stating through the optimization of InfoNCE Loss, specifically $\mathcal{L}_{align}$, encoder Lipschitz constant is effectively upper-bounded.

**Theorem 2.** *Assume the measure of the support intersection $S = \text{supp}(\pi, p) \cap \text{supp}(\tilde{\pi}, \tilde{p})$ satisfies $\mathbb{P}(S) > \alpha$ for some $\alpha > 0$, then $L_E$ is strictly upper-bounded by the square root of $\mathcal{L}_{align}$.*

Refer to Appendix A.6 for a complete proof. Intuitively, the radius of latent cluster produced under dynamics $p$ can be bounded with $\mathcal{L}_{align}$. We then prove condition $\mathbb{P}(S) > \alpha$ yields a "bridge sample" among marginally different dynamics (as shown in Fig. 1(b)), through which the distance between centroids can be bounded by two times the cluster radius. Although this upper-bound only relates to $\mathcal{L}_{align}$, simply minimizing the Alignment Loss yields an easy local-optimum: mapping all trajectory segments to the same latent, in which case the latent space becomes meaningless. We argue that it is critical to simultaneously enforce both *smoothness* ($\mathcal{L}_{align}$) and *structure* ($\mathcal{L}_{uniform}$), a claim we support via experiments in Subsection 5.4.

From a practical perspective, InfoNCE Loss helps encoder focus on distinguishing trajectories using dynamics factors rather than nuisance variables, as stated in the following theorem. This enhancement is crucial for dynamics-related latent space learning where information distillation from long time windows is typical, causing pure reconstructive methods to fail.

**Theorem 3.** *Let the trajectory space $\mathcal{T}$ be locally factorizable into dynamics-relevant features $\mathcal{D}$ and nuisance features $\mathcal{S}$, such that trajectory $\tau \approx (\mu, s)$. Then minimizing the InfoNCE Loss (Eq. (9)) implies minimizing the Frobenius norm of $\partial E/\partial s$.*

Proof is presented in Appendix A.6. This result parallels the motivation in vision-based reinforcement learning, where robust representations should remain invariant to task-irrelevant visual variations (Zhang et al., 2021; Liu

et al., 2023). Our theorem here conveys the same principle, but instead of relying on reward-based supervision as in prior work, it shows that task-relevant structure can emerge in a semi-supervised manner through contrastive learning.

### 4.3. Practical Algorithm

We now formalize framework into a tractable learning procedure. Instead of projecting explicit dynamics parameters to latent vectors, we infer a latent probabilistic context $z$ via a variational information bottleneck. We map a history of interactions to this latent space, maximizing the Evidence Lower Bound (ELBO (Kingma & Welling, 2014; Higgins et al., 2017a)) on $p(s_{t+1}|s_t, a_t)$:

$$
\begin{aligned}
\mathcal{L}_{\text{VAE}} = - \; & \mathbb{E}_{q_\phi(z_t|w_t)} \left[ \log p_\theta(s_{t+1}|s_t, a_t, z_t) \right] \\
& + \beta D_{KL} \left( q_\phi(z_t|w_t) \| p(z_t) \right)
\end{aligned}
\tag{10}
$$

where $q_\phi$ is an amortized inference network mapping history windows $w_t$ to the latent distribution, and $p_\theta$ is the generative decoder. We impose an isotropic Gaussian prior $p(z_t) = \mathcal{N}(0, \mathbf{I})$. Let $\mathcal{B}$ denote a minibatch of trajectory segments, define the set of indices $\mathcal{P}(i)$ as the positive set for an anchor segment $i$, containing all segments $j \in \mathcal{B} \backslash \{i\}$ generated under the same dynamics parameters $\mu_i$:

$$
\mathcal{P}(i) = \{ j \in \mathcal{B} \setminus \{i\} \mid \mu_j = \mu_i \}
\tag{11}
$$

Multi-Positive InfoNCE Loss (Khosla et al., 2020) is then employed to enforce a small $L_E$ (Theorem 2), as well as ensuring that the learned embedding captures the underlying physical properties rather than nuisance variables:

$$
\mathcal{L}_{\text{contrast}} = \frac{1}{|\mathcal{B}|} \sum_{\substack{i \in \mathcal{B} \\ |\mathcal{P}(i)| > 0}} \frac{1}{|\mathcal{P}(i)|} \sum_{j \in \mathcal{P}(i)} \ell_{i,j}
\tag{12}
$$

This objective provides three complementary training signals (effect visualized in Fig. 1(b)): (1) *Temporal Consistency:* forcing sub-sequences from the same trajectory (e.g., the blue cluster) to map to proximate latent points, which quantitatively minimizes the local consistency metric $\delta_e$; (2) *Manifold Continuity:* utilizing "bridge samples" with marginally different dynamics that share partial overlap in behavior to pull similar clusters closer in the embedding space, thus bounding $L_E$ (as established in proof of Theorem 2); and (3) *Global Structure:* using the uniformity component to ensure that the latent manifold maximally spans the available space (e.g., the blue vs. green cluster), thus forming global structure.

The total training objective combines task performance with these representation learning auxiliary losses:

$$
\mathcal{L} = \mathcal{L}_{\text{rl}} + \lambda_1 \mathcal{L}_{\text{VAE}} + \lambda_2 \mathcal{L}_{\text{contrast}}
\tag{13}
$$

where $\mathcal{L}_{\text{rl}}$ is the reinforcement learning loss. Since policy is conditioned on $z$, $\mathcal{L}_{\text{rl}}$ also flows back to the encoder, en-

couraging the output of task-relevant latents. Pseudocode for our algorithm is provided in Appendix B.1.

## 5. Experiments

We evaluate our method on four MuJoCo continuous-control benchmarks (Todorov et al., 2012): *Hopper*, *Walker2d*, *HalfCheetah*, and *Ant*, covering diverse locomotion challenges (e.g., Hopper's flight phase, planar locomotion for Walker2d/HalfCheetah, and 3D coordination for Ant). To induce dynamics variations, we randomize four physical properties: body mass, joint damping, slide friction, and torque scale. Each is applied as a multiplicative scalar across relevant body parts (e.g., mass scale 0.5 halves all link masses), yielding an 8–36D randomized dynamics space depending on morphology. Refer to Appendix B.2 for parameter ranges and per-environment dimensions.

Our method is agnostic to reinforcement learning algorithm, but we use Soft Actor-Critic (SAC (Haarnoja et al., 2018)) for its sample and exploration efficiency. We compare the proposed method with five baselines: (1) *SAC+DR*, SAC with dynamics randomization (Peng et al., 2018); (2) *RMA (Phase 1)*, an oracle that conditions on ground-truth dynamics parameters (i.e., system identification) (Kumar et al., 2021); (3) *RMA (Phase 2)*, employing an encoder to regress the Phase-1 latent from trajectory history (Kumar et al., 2021); (4) *SO-CMA*, CMA-ES (Hansen et al., 2003) search in the RMA latent space for $z$ maximizing target return (Yu et al., 2019); and (5) *VAE*, our variational ablation trained without $\mathcal{L}_{\text{contrast}}$. The source code of LDG along with baselines are available online[1].

### 5.1. In-Distribution Stability

We evaluate in-distribution stability by testing asymptotic performance on 5 sets of dynamics parameters sampled within the training range (Table 1). LDG achieves the highest mean reward on Hopper (3239.2) and Walker2d (4883.7), and ranks second on Ant, close to the test-time oracle SO-CMA (which uses $\approx 42k$ samples per setting). LDG also yields substantially lower variance across seeds (e.g., Walker2d: $\sigma = 381.6$ vs. $\sigma = 1382.3$ for RMA (Phase 2)), an expected stability consistent with Theorem 1 and: contrastive learning upper-bounds $L_E$, reducing term (III) in Eq. (7). In contrast, RMA (Phase 2) can exhibit high-variance when the trajectory-to-parameter mapping is ill-posed (e.g., Hopper's flight phase), and the VAE ablation suffers from latent irregularity; enforcing local consistency (small $\delta_e$ and $L_E$) keeps nearby histories mapped to a compact latent region and provides a stable condition-

---

[1]https://github.com/Mr-Wonderfool/Latent-Dynamics-Geometries

*Table 1.* Comparison of asymptotic in-distribution performance. Results report the mean cumulative reward $\pm$ one standard deviation over 5 sets of dynamics parameters. Bold and underline indicate top-1 and top-2 performance within the same access regime (excluding test-time optimization methods).

| | Test-time opt. | Source-only + Zero-shot | | | | |
|---|---|---|---|---|---|---|
| Env | SO-CMA | SAC+DR | RMA (Phase 1) | RMA (Phase 2) | VAE | LDG (Ours) |
| Hopper | $635.2 \pm 489.9$ | $1767.3 \pm 1183.2$ | $1799.6 \pm 1108.0$ | $1677.5 \pm 1168.2$ | $\underline{2029.6 \pm 819.1}$ | $\mathbf{3239.2 \pm 279.3}$ |
| Walker2d | $2281.9 \pm 1336.7$ | $3170.2 \pm 162.0$ | $3132.4 \pm 1126.5$ | $\underline{3193.6 \pm 1382.3}$ | $2462.4 \pm 1126.0$ | $\mathbf{4883.7 \pm 381.6}$ |
| HalfCheetah | $4066.3 \pm 989.0$ | $4265.9 \pm 825.5$ | $\mathbf{6713.3 \pm 1521.6}$ | $\underline{6635.5 \pm 1433.5}$ | $5127.6 \pm 850.4$ | $5799.1 \pm 1053.4$ |
| Ant | $5042.9 \pm 494.3$ | $\underline{4729.8 \pm 226.3}$ | $3817.7 \pm 1512.4$ | $4107.8 \pm 1610.9$ | $2887.9 \pm 534.7$ | $\mathbf{5183.9 \pm 296.9}$ |

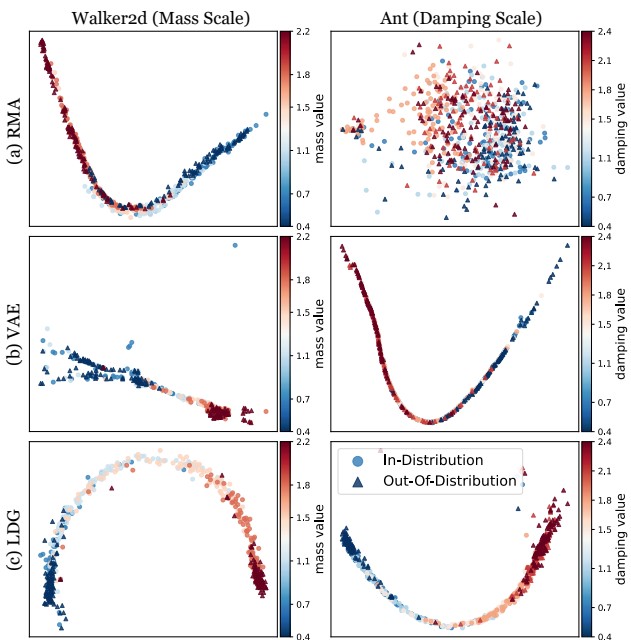

*Figure 2.* T-SNE visualization of the latent structure. In Walker2d and Ant, mass and damping scale is varied respectively, with range covering both in-distribution data (marked as circles) and out-of-distribution data (marked as triangles). (a) RMA (Phase 2) produces a scattered embedding with no clear ordering and cluster boundary. (b) VAE suffers from mode collapse or topological disjointedness. (c) LDG (Ours) uncovers a smooth, monotonic arc where latent coordinates correlate linearly with physical parameters. This ordered geometry enables the encoder to extrapolate OOD dynamics (triangles) by extending the manifold structure learned from ID data.

ing signal. LDG underperforms RMA on HalfCheetah; we hypothesize the imposed stability constraint may over-regularize in highly reactive gaits that require large, immediate force commands. We further analyze this problem within "failure mode analysis" in subsection 5.2.

This stability is reflected in the learned latent geometry. Fig. 2 illustrates the embedding space for Walker2d and Ant under controlled variations of a single physical parameter (mass and damping respectively), while keeping others fixed. The baseline methods, particularly RMA (Phase 2),

produce scattered and unstructured latents, explaining the high variance in Table 1: without a consistent mapping, small noise in the input trajectory can lead to widely different latent codes and destabilize the policy. The VAE ablation also suffers from topological disconnectedness, failing to capture the global continuum of the physical property. In contrast, LDG discovers a smooth, monotonic manifold (an arc) where clusters induced by similar dynamics are adjacent, providing empirical support for our analysis: optimizing InfoNCE keeps similar dynamics close and preserves parameter continuity.

While $\mathcal{L}_{align}$ alone can upper-bound the encoder's Lipschitz constant in theory, we find $\mathcal{L}_{uniform}$ necessary in practice to separate clusters and avoid mode collapse, yielding the continuous arc in Fig. 2(c). This ordered geometry also facilitates extrapolation: OOD dynamics (triangles) extend the learned manifold, indicating *perception-level zero-shot generalization* as a prerequisite for control-level zero-shot generalization.

### 5.2. Zero-Shot Generalization

We evaluate zero-shot adaptation in three regimes: (1) *Un-modeled Parameter*, where a property (e.g., mass or damping) takes on OOD values at test time (only $1.0\times$ is in-distribution); (2) *Time-Varying Dynamics* (*Var. Env*), where a factor in $[0.9, 1.1]$ rescales dynamics parameters every 200 steps within an episode; and (3) *Structural Failure* (*Struct. Fail.*), where one actuator is disabled (command set to zero) mid-episode after 200 environment steps (allowing methods such as LDG and phase-2 RMA to infer a stable latent representation prior to the structural perturbation), causing a catastrophic shift not representable by continuous dynamics parameters. Results are summarized in Table 2, where rewards are averaged over 5 random seeds. For structural failure setting, we only record reward after the broken joint event. We train all weights using the randomization ranges in Appendix Table B.2.

**Robustness to Structural Shifts.** Failure of SO-CMA and RMA in the *Struct. Fail.* scenario reveals a critical limitation of explicit identification methods. Since these base-

*Table 2.* Zero-shot generalization performance across three OOD settings: (1) *Unmodeled Parameter* (columns 1-3), where specific physical properties not randomized during training are tested in target domain; (2) *Time-Varying Dynamics* (*Var. Env*), where dynamics parameters shift mildly every 200 time steps; and (3) *Structural Failure* (*Struct. Fail.*), where a joint failure is introduced mid-episode through zero-command enforcement after some delay. Bold and underline indicate top-1 and top-2 performance within the same access regime (excluding test-time optimization methods).

| Method | Hopper (Mass Scale) | | | | | Walker2d (Damping Scale) | | | | |
|---|---|---|---|---|---|---|---|---|---|---|
| | 0.5× | 1.0× | 2.0× | Var. Env | Struct. Fail. | 0.3× | 1.0× | 2.2× | Var. Env | Struct. Fail. |
| **Test-time Optimization:** | | | | | | | | | | |
| SO-CMA | 1121 | 3260 | 402 | 420 | 432 | 4214 | 5193 | 271 | 649 | 504 |
| **Source-only + Zero-shot:** | | | | | | | | | | |
| SAC+DR | 778 | 3011 | 809 | 2701 | 182 | 4734 | 4696 | 4651 | 3351 | 739 |
| RMA (Phase 1) | 625 | 2545 | **2747** | 2192 | 149 | 3667 | 3389 | 4237 | 3121 | 364 |
| RMA (Phase 2) | 3092 | 3102 | 563 | 3089 | 394 | 4117 | 4139 | 4096 | 4062 | 292 |
| VAE | 2810 | 2893 | 711 | 2628 | 255 | 4733 | 4749 | 4708 | 4673 | 238 |
| **LDG (Ours)** | **3154** | **3434** | 1349 | **3294** | **596** | **5342** | **5320** | **5349** | **5293** | **952** |

| Method | HalfCheetah (Mass Scale) | | | | | Ant (Damping Scale) | | | | |
|---|---|---|---|---|---|---|---|---|---|---|
| | 0.5× | 1.0× | 2.0× | Var. Env | Struct. Fail. | 0.3× | 1.0× | 2.2× | Var. Env | Struct. Fail. |
| **Test-time Optimization:** | | | | | | | | | | |
| SO-CMA | 5442 | 9602 | 2965 | 5574 | 2542 | 4131 | 4059 | 4963 | 4787 | 356 |
| **Source-only + Zero-shot:** | | | | | | | | | | |
| SAC+DR | 5375 | 5753 | 3705 | 5769 | 1593 | 3443 | 3909 | 4228 | 2044 | **1533** |
| RMA (Phase 1) | 6542 | **9589** | **5676** | **9407** | **2913** | 2011 | 3182 | 3122 | 2290 | -7 |
| RMA (Phase 2) | **7123** | 9281 | 5615 | 9121 | 2862 | 1567 | 1033 | 666 | 623 | 72 |
| VAE | 6421 | 6466 | 4566 | 6399 | 2291 | 3629 | 2062 | 3602 | 3087 | 729 |
| **LDG (Ours)** | 5999 | 8022 | 5284 | 7849 | 2071 | **4797** | **4804** | **4789** | **3462** | 1003 |

lines are trained to regress or search for specific latent vectors that correspond to some dynamics parameters, their latent vocabulary is constrained to the manifold of valid physics simulations. A "broken joint" is an OOD event that does not correspond to any valid combination of dynamics parameters, causing SO-CMA to fail catastrophically as it attempts to locate a non-existent latent. In contrast, LDG learns a functional manifold of trajectory dynamics; because the broken joint produces a trajectory pattern similar to extremely high damping and mass on this joint, encoder can project it to a valid, albeit extrapolative, region of the latent space, enabling adaptation.

**Adaptation to Non-Stationarity.** In the *Var. Env* setting, RMA (Phase 2) often underperforms LDG (e.g., Ant: 623 vs. 3462). When dynamics switch mid-window, the input trajectory contains conflicting physical evidence, leading RMA to regress an average and less representative latent. LDG, however, is trained to enforce local consistency. A window containing a dynamics switch is simply treated as a transition between two latent clusters. The encoder is robust to these intermediate states, allowing the policy to transition smoothly between behavioral modes without being constrained to a single global identification.

**Failure Mode Analysis.** Despite these successes, we observe that LDG underperforms baselines on the HalfCheetah environment (for *Var. Env* and *Struct. Fail.*). HalfCheetah requires fast, high-frequency gait cycles where optimal policies are often highly reactive, exerting large, immediate forces. Visualizations of HalfCheetah latent space (Fig. 3(a)) show that under variations of a single dynamics parameter, latent clusters share little overlapping regions. This indicates that latent distances scale disproportionately with dynamics differences, hindering the formation of "bridge samples" that leads to robust adaptation. We hypothesize that in such highly dynamic environments, the Lipschitz smoothness constraint imposed by our contrastive objective may induce an over-regularization effect. By penalizing sharp transitions in the latent space, the method might dampen the aggressive, high-frequency adaptation required for maximum velocity in HalfCheetah, favoring stable locomotion over the explosive reactivity exploited by unconstrained baselines like RMA.

**Outcome-Centric vs. Parameter-Centric Adaptation.** RMA Phase 2 (learned adaptation) often yields higher returns compared with RMA Phase 1 (oracle parameters) (e.g., Walker2d: 4930 vs 3389). This corroborates our

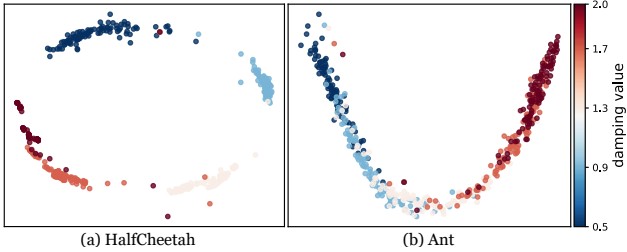

(a) HalfCheetah       (b) Ant

*Figure 3.* T-SNE visualization of implicit structure discovery. The encoder was trained on environments where joint damping was held fixed, yet it successfully organizes unseen damping variations into a coherent, ordered manifold during testing. This indicates that LDG learns functional properties (i.e., resistance to motion) rather than specific parameter labels, allowing it to generalize to unseen physical properties that induce similar dynamic effects.

design choice to condition on interaction history rather than explicit parameters. The "ground truth" parameters are static labels that do not capture the immediate interactions between the robot and its environment (e.g., a foot slipping). A trajectory-based encoder (used in both RMA Phase 2 and LDG) can react to these instantaneous interaction forces, effectively allowing the agent to adapt to the manifestation of dynamics rather than just their underlying values, as analyzed during method development. This observation is consistent with the student-teacher phenomenon reported in reinforcement learning literatures, where a student policy can surpass the performance of its teacher after distillation or imitation (Rusu et al., 2015; Lee et al., 2020a).

### 5.3. Structure Discovery Via Functional Equivalence

LDG holds the ability to discover the structure of physical parameters even when they are not explicitly randomized during training. As shown in Fig. 3, when training an agent on randomized mass but keeping joint damping fixed, the learned encoder still organizes damping values into a coherent, ordered manifold during testing. This can be attributed to *functional equivalence*: high joint damping and increased mass all manifest as "resistance to motion", leading to the discovery of "resistance" manifold through LDG optimization. When LDG encounters unseen damping variations, the encoder projects them onto this pre-existing manifold based on their functional effect. This explains the robustness observed in the OOD experiments: the method does not need to identify the exact physical label (e.g., "broken joint"), but rather maps the resulting trajectory dynamics to a known functional coordinate (e.g., "extreme resistance"). However, this transfer is not universal; it relies on the training distribution containing sufficient functional variance. Empirically we find if the training parameters (e.g., high damping) exert a significantly weaker influence on dynamics than the unseen

test parameter (e.g., high mass), the learned manifold is too compressed to represent the new, larger variations.

### 5.4. Alternative Regularizer

Theorem 1 establishes that reducing the encoder's Lipschitz constant, $L_E$, minimizes target domain regret. While we enforce this geometric smoothness via contrastive learning, a natural alternative is Spectral Normalization (Yoshida & Miyato, 2017; Miyato et al., 2018), a technique widely proven to improve the robustness of generative models. By dividing each weight matrix by its spectral norm, SN explicitly bounds the Lipschitz constant of each layer, thereby strictly upper-bounding $L_E$ by 1.

To isolate the structural benefits of contrastive learning, we ablate our objective by replacing it with SN applied directly to the encoder layers. We evaluate the resulting policies on the Walker2d and Ant environments across In-Distribution (ID), Out-Of-Distribution (OOD) damping, time-varying (Var. Env), and structural failure (Struct. Fail) scenarios. The results are summarized in Table 3, rewards are averaged over 5 random seeds. While SN achieves strong ID performance on Ant environment—suggesting that explicit spectral constraints successfully minimize $L_E$ to ensure local stability—it struggles significantly in OOD and structural failure scenarios across both domains. We attribute this degradation to SN's inability to enforce a meaningful latent topological structure. Although SN mathematically ensures encoder smoothness, it does not organize the latent manifold based on physical semantics. In contrast, contrastive learning guarantees this organized geometry through the uniformity loss term in the InfoNCE decomposition, a property that proves critical for the policy to successfully extrapolate to unseen physical variations in zero-shot settings. This ablation result indicates that advantage of contrastive learning stems from both smoothness ($\mathcal{L}_{align}$) and structure ($\mathcal{L}_{uniform}$), though theoretically only $\mathcal{L}_{align}$ is needed to upper-bound $L_E$.

### 5.5. Hyperparameter Sensitivity

Overall, LDG is robust to hyperparameters and does not require aggressive tuning. We found LDG only sensitive to reward scale (tuned to balance $\mathcal{L}_{rl}$ in Eq. (13)), VAE encoder $\beta$ (in Eq. (10)), encoder horizon $H$ and dimension of latent vectors $d_z$. We provide tuning heuristics for reward scale and $\beta$ in Appendix C, and perform an ablation in Fig. 4 to guide the selection of $H$ and $d_z$.

**Latent Dimension Capacity** ($d_z$). As shown in Fig. 4(a) and (c), we observe a clear trade-off between information bottlenecking and optimization complexity. Empirically, the minimum effective ratio between latent and observation dimensions is approximately $3 : 16$; below which the latent code is largely ignored relative to the observa-

*Table 3.* Performance comparison of Contrastive Learning (CL) versus Spectral Normalization (SN). Rewards are averaged over 5 random seeds.

| Env | Method | ID | OOD Damping 0.3× | OOD Damping 2.2× | Var. Env | Struct. Fail |
|---|---|---|---|---|---|---|
| **Walker2d** | SN | 4384 | 4646 | 4677 | 4661 | 335 |
| | CL (Ours) | **4884** | **5342** | **5349** | **5293** | **952** |
| **Ant** | SN | **5538** | 3329 | 2612 | 2969 | 248 |
| | CL (Ours) | 5184 | **4797** | **4789** | **3462** | **1003** |

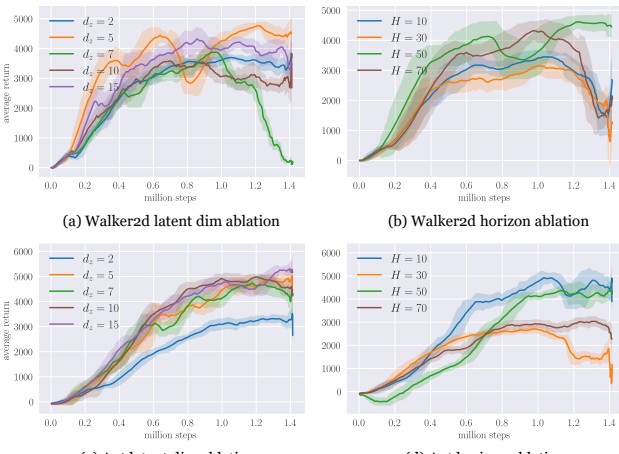

(a) Walker2d latent dim ablation     (b) Walker2d horizon ablation

(c) Ant latent dim ablation     (d) Ant horizon ablation

*Figure 4.* Training reward curves on Walker2d and Ant environments. Subfigures (a) and (c) illustrate the sensitivity to the latent dimension ($d_z$), highlighting the tradeoff between representational bottlenecking and manifold sparsity. Subfigures (b) and (d) demonstrate the effect of trajectory horizon length ($H$), balancing temporal context accumulation against immediate control reactivity. Solid lines represent the mean episode return, and shaded regions denote the variance across environments.

tions, as evidenced by the failure of $d_z = 2$ in Walker2d and Ant. Conversely, optimization over high-dimensional Gaussian latent spaces is known to be challenging, and excessive latent capacity ($d_z \in \{7, 10, 15\}$) introduces sparsity that destabilizes training in Walker2d. In contrast, the higher-dimensional Ant environment remains relatively insensitive to latent dimensions larger than $5$, yielding comparable returns across settings.

**Temporal Context Horizon ($H$).** The choice of horizon length must balance contextual information accumulation against control reactivity. For a fair comparison, trajectory windows are compressed to the same output feature dimension by adjusting the stride and kernel size of the convolutional encoder accordingly. Extremely short horizons (e.g., $H = 10$) provide insufficient historical context for stable latent inference, leading to suboptimal or highly volatile training dynamics. In contrast, excessively long

horizons ($H = 70$) dilute the importance of recent transitions. Across both environments, $H = 50$ consistently yields the best performance. Additional analysis on responsiveness under dynamics shifts for different context horizon is provided in Appendix C.

## 6. Conclusion

We introduce Latent Dynamics Geometries (LDG), a framework for robust zero-shot policy adaptation that moves beyond RMA-style system identification by learning a geometrically structured latent dynamics manifold. We link target-domain regret to the Lipschitz smoothness of a trajectory encoder, and instantiate this idea with a practical algorithm that uses contrastive learning to control representation sensitivity. Empirically, LDG improves in-distribution stability and achieves strong zero-shot generalization across diverse dynamics shifts, often matching or surpassing parameter-centric (RMA) and unstructured variational (VAE) baselines, while producing a more ordered latent geometry. Our results further suggest that LDG can exploit functional equivalences among physical factors to handle certain unmodeled events by mapping them to meaningful regions of the learned manifold.

Empirical validations also reveal a fundamental trade-off: enforcing smoothness for robust adaptation may over-regularize highly reactive tasks such as HalfCheetah. Future work will investigate mechanisms for adaptively relaxing smoothness constraints when necessary. In addition, we aim to move toward a fully unsupervised setting that removes the need to explicitly determine whether dynamics parameters are shared across samples, and to extend LDG to more complex domains, including interactive tasks.

## Acknowledgements

We would like to thank Zihan Xu for insightful discussions and Haodong Yang for helping setup the initial codebase. This work was supported by the National Natural Science Foundation of China under Grants 62403358, 62333017

and 62233013.

## Impact Statement

This paper presents work whose goal is to advance the field of Machine Learning. There are many potential societal consequences of our work, none of which we feel must be specifically highlighted here.

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

# A. Proof of Lemmas And Theorems

## A.1. Notations And Definitions

Throughout this section, we use subscript $T$ to denote quantities in the target domain, and use subscript $S$ to denote quantities in the source domain, such that $\pi_S$ denotes an adaptive policy in the source domain, and $V_S^{\pi_S}$ denotes the value function achieved by rolling out $\pi_S$ in the source domain. For notational convenience we assume the state space is discrete; it is straightforward to extend all of our proofs to continuous domains by replacing summations with integrals.

### A.1.1. DYNAMICS IN MDPS

We consider a family of Markov Decision Processes parameterized by physical parameters $\theta \in \Theta$:

$$\mathcal{M}(\theta) := (\mathcal{S}, \mathcal{A}, p_\theta, r, \gamma) \tag{A.1}$$

where the state space $\mathcal{S}$, action space $\mathcal{A}$, reward function $r$, and discount factor $\gamma$ are shared across domains, while the transition dynamics $p_\theta(s'|s, a)$ depend on the underlying system parameters $\theta$ (e.g., mass distribution, actuation characteristics, or contact properties). We make the following assumptions regarding dynamics parameters and their induced transition functions.

**Assumption A.1** (Dynamics Continuity in Parameters)**.** There exists a constant $L_p > 0$ such that for any $\theta, \theta' \in \Theta$ and $\forall(s, a)$, $\|p_\theta(\cdot|s, a) - p_{\theta'}(\cdot|s, a)\|_1 \le L_p \|\theta - \theta'\|$.

**Definition A.1** (Task-Relevant Dynamics Equivalence)**.** Let $\theta \in \Theta$ denote the underlying physical parameters of a system. We define a task-relevant dynamics representation $\eta := \psi(\theta) \in \mathcal{H}$, where $\psi$ maps system parameters to a low-dimensional space capturing task-relevant interaction properties. Two parameter settings $\theta$ and $\theta'$ are said to be *dynamics-equivalent* if $\psi(\theta) = \psi(\theta')$. We denote the corresponding equivalence class by

$$[\theta] := \{\theta' \in \Theta \mid \psi(\theta') = \psi(\theta)\} \tag{A.2}$$

**Assumption A.2** (Equivalence-Class-Induced Dynamics)**.** The transition dynamics depend on $\theta$ only through its task-relevant representation $\eta = \psi(\theta)$, i.e., $p_\theta(s'|s, a) \approx p_\eta(s'|s, a)$. Consequently, dynamics parameters within the same equivalence class $[\theta]$ induce approximately identical trajectory distributions for the task of interest.

Therefore, the qualitative claim of "different set of dynamics parameters can induce similar dynamics" essentially means equivalence class $[\theta] \ne \emptyset$, and our algorithm learns function $\psi$ to identify equivalence classes.

### A.1.2. WINDOW DISTRIBUTION

For a window segment $w_t = (s_{t-H}, a_{t-H}, \ldots, s_{t-1}, a_{t-1})$ of one full trajectory $\tau = (s_0, a_0, \ldots, s_{L-1}, a_{L-1}, s_L)$, its marginal density $\rho(\pi, p)$ can be obtained by rolling out the Markov chain:

$$\rho_t(\pi, p) := \rho(\pi, p)(w_t) = d_{t-H}^{\pi, p}(s_{t-H}) \cdot \prod_{k=t-H}^{t-1} \pi(a_k|s_k, E(w_k)) \cdot p(s_{k+1}|s_k, a_k) \tag{A.3}$$

where $d_{t-H}^{\pi, p}(s_{t-H})$ is the probability density of being in state $s_{t-H}$ at time step $t - H$ under policy $\pi$ and dynamics $p$, defined recursively as:

$$d_k^{\pi, p}(s_k) = \int_\mathcal{S} \int_\mathcal{A} p(s_k|s_{k-1}, a_{k-1})\pi(a_{k-1}|s_{k-1}, E(w_{k-1}))d_{k-1}^{\pi, p}(s_{k-1}) \, da_{k-1} \, ds_{k-1} \tag{A.4}$$

In subsequent analysis, we will write $E(w_k)$ as $z_k$ for brevity.

### A.1.3. DEFINITIONS

The following definitions are established in the main thesis, we restate it here to aid the proof of theorems.

**Definition A.2.** Two MDPs $\mathcal{M}_S$ and $\mathcal{M}_T$ are $\epsilon_p$-close in dynamics, if they share the same state space, action space and reward function, but differ in transition dynamics $p_S(s'|s, a)$ and $p_T(s'|s, a)$, such that:

$$D_{TV}(p_T(\cdot|s, a), p_S(\cdot|s, a)) \le \epsilon_p \quad \forall(s, a) \tag{A.5}$$

**Definition A.3.** A latent-conditioned policy $\pi(\cdot|s, z)$ is $L_\pi$-smooth, if there exists a constant $L_\pi$ such that:

$$\sup_s D_{TV}(\pi(\cdot|s, z_S), \pi(\cdot|s, z_T)) \leq L_\pi \|z_S - z_T\|_2 \tag{A.6}$$

**Definition A.4.** For a fixed policy $\pi$ and transition dynamics $p$, the time-indexed latent centroid $\mu_t(\pi, p)$ is the expected encoding of the window at that specific time step:

$$\mu_t(\pi, p) = \mathbb{E}_{w \sim \rho_t(\pi, p)}[E(w)] \tag{A.7}$$

**Definition A.5.** An encoder is $\delta_e$-consistent if:

$$\sup_t \sup_{w \sim \rho_t(\pi, p)} \|E(w) - \mu_t(\pi, p)\|_2 \leq \delta_e \tag{A.8}$$

**Definition A.6.** The encoder Lipschitz constant $L_E$ with respect to the total variation divergence of the induced window distributions is defined as:

$$L_E = \sup_t \sup_{(\pi, p), (\tilde{\pi}, \tilde{p})} \frac{\|\mu_t(\pi, p) - \mu_t(\tilde{\pi}, \tilde{p})\|_2}{D_{TV}(\rho_t(\pi, p), \rho_t(\tilde{\pi}, \tilde{p}))} \tag{A.9}$$

The encoder is considered $L_E$-smooth if $L_E < \infty$.

**Definition A.7.** Let $\mathcal{W}$ be the space of trajectory windows, the support of physically feasible windows under dynamics $p$ and policy $\pi$ is defined as:

$$\text{supp}(\pi, p) = \{w \in \mathcal{W} \mid \forall(s, a, s') \in w, p(s'|s, a) > 0\} \tag{A.10}$$

## A.2. Simulation Lemma

Below we present the Simulation Lemma (Kearns & Singh, 2002), which bounds the difference in cumulative reward achieved by the same (nonadaptive) policy under different domains.

**Lemma A.1** (Simulation Lemma (Kearns & Singh, 2002))**.** *Let $\mathcal{M}_S$ and $\mathcal{M}_T$ be two MDPs that are $\epsilon_p$-close in dynamics, then for policy $\pi(\cdot|s)$, the following bound holds:*

$$\|V_T^\pi - V_S^\pi\|_\infty \leq \frac{2\gamma R_{\max}}{(1 - \gamma)^2} \epsilon_p \tag{A.11}$$

*Proof.* In subsequent proof we drop the explicit dependency on action in the transition function for brevity, to write $p_T(s'|s, \pi(a|s))$ as $p_T(s'|s)$. Recall the Bellman equation for a fixed policy $\pi$:

$$V_T^\pi(s) = R(s) + \gamma \sum_{s'} p_T(s'|s) V_T^\pi(s') \tag{A.12}$$

Denote the value difference $V_T^\pi(s) - V_S^\pi(s)$ as $\Delta(s)$, which can be expanded as:

$$
\begin{aligned}
\Delta(s) &= \gamma \left( \sum_{s'} p_T(s'|s) V_T^\pi(s') - \sum_{s'} p_S(s'|s) V_S^\pi(s') \right) \\
&= \gamma \left( \sum_{s'} p_T(s'|s) V_T^\pi(s') - \sum_{s'} p_T(s'|s) V_S^\pi(s') + \sum_{s'} p_T(s'|s) V_S^\pi(s') - \sum_{s'} p_S(s'|s) V_S^\pi(s') \right) \\
&= \gamma \underbrace{\sum_{s'} p_T(s'|s) \Delta(s')}_{\text{Value Error Propagation}} + \gamma \underbrace{\sum_{s'} (p_T(s'|s) - p_S(s'|s)) V_S^\pi(s')}_{\text{Dynamics Error}}
\end{aligned} \tag{A.13}
$$

The Dynamics Error term can be bounded by Hölder's inequality:

$$\left| \sum_{s'} (p_T(s'|s) - p_S(s'|s)) V_S^\pi(s') \right| \leq \|p_T(\cdot|s) - p_S(\cdot|s)\|_1 \|V_S^\pi\|_\infty \leq 2\epsilon_p \frac{R_{\max}}{1 - \gamma} \tag{A.14}$$

Substituting this back into Eq. (A.13), taking the absolute value and utilizing triangle inequality:

$$|\Delta(s)| \le \gamma \left| \sum_{s'} p_T(s'|s)\Delta(s') \right| + \frac{\gamma R_{\max}}{1-\gamma}\epsilon_p \le \gamma \max_{s'}|\Delta(s')| + \frac{\gamma R_{\max}}{1-\gamma}\epsilon_p \tag{A.15}$$

We further take the max operator on both sides of Eq. (A.15), which leads to:

$$\max|\Delta(s)| = \|\Delta(s)\|_\infty \le \gamma\|\Delta(s)\|_\infty + \frac{\gamma R_{\max}}{1-\gamma}\epsilon_p \tag{A.16}$$

Rearranging terms, we obtain the bound in Eq. (A.11). $\qquad\square$

### A.3. Extended Performance Difference Lemma

Standard Performance Difference Lemma (PDL (Kakade & Langford, 2002)) focuses on value function difference induced by different policies within the same domain. Since different adaptive policies naturally require different domains, we extend PDL to bound $V_T^{\pi_T} - V_S^{\pi_S}$.

**Lemma A.2** (Extended Performance Difference Lemma). *For policy and dynamics pairs $(\pi_T, p_T)$ in target domain and $(\pi_S, p_S)$ in source domain, starting from the same initial state $s_0$, policy performance difference is bounded by:*

$$|V_T^{\pi_T}(s_0) - V_S^{\pi_S}(s_0)| \le 2R_{max}\sum_{t=0}^{\infty}\gamma^t \left( D_{TV}(d_t^{\pi_S,p_S}, d_t^{\pi_T,p_T}) + \mathbb{E}_{s_t \sim d_t^{\pi_S,p_S}}\left[ D_{TV}(\pi_S(\cdot|s_t, z_{S,t}), \pi_T(\cdot|s_t, z_{T,t})) \right] \right) \tag{A.17}$$

*where $d_t^{\pi,p}$ denotes the state marginal at time $t$, induced by policy $\pi$ and dynamics $p$, defined in Eq. (A.4).*

*Proof.* By definition, the expected return of a policy from an initial state $s_0$ is the sum of expected rewards over the joint state-action marginals at each time step $t$:

$$\begin{aligned} V_S^{\pi_S}(s_0) &= \sum_{t=0}^{\infty}\gamma^t \int_{\mathcal{S}}\int_{\mathcal{A}} \xi_{S,t}(s_t, a_t)R(s_t, a_t)\, da_t\, ds_t \\ V_T^{\pi_T}(s_0) &= \sum_{t=0}^{\infty}\gamma^t \int_{\mathcal{S}}\int_{\mathcal{A}} \xi_{T,t}(s_t, a_t)R(s_t, a_t)\, da_t\, ds_t \end{aligned} \tag{A.18}$$

where $\xi_{S,t}(s_t, a_t) = d_t^{\pi_S,p_S}(s_t)\pi_S(a_t|s_t, z_{S,t})$. Evaluate the absolute value of performance difference and apply the triangle inequality:

$$\begin{aligned} |V_S^{\pi_S}(s_0) - V_T^{\pi_T}(s_0)| &= \left| \sum_{t=0}^{\infty}\gamma^t \int_{\mathcal{S}}\int_{\mathcal{A}} (\xi_{S,t}(s_t, a_t) - \xi_{T,t}(s_t, a_t))R(s_t, a_t)\, da_t\, ds_t \right| \\ &\le 2R_{max}\sum_{t=0}^{\infty}\gamma^t D_{TV}(\xi_{S,t}(s_t, a_t), \xi_{T,t}(s_t, a_t)) \end{aligned} \tag{A.19}$$

Apply the triangle inequality again, we split the Total Variation divergence into two integrals:

$$\begin{aligned} D_{TV}(\xi_{S,t}(s_t, a_t), \xi_{T,t}(s_t, a_t)) &\le \frac{1}{2}\int_{\mathcal{S}}\int_{\mathcal{A}} |d_t^{\pi_S,p_S}(s_t)|\,|\pi_S(a_t|s_t, z_{S,t}) - \pi_T(a_t|s_t, z_{T,t})|\, da_t\, ds_t \\ &\quad + \frac{1}{2}\int_{\mathcal{S}}\int_{\mathcal{A}} |\pi_T(a_t|s_t, z_{T,t})|\,|d_t^{\pi_S,p_S}(s_t) - d_t^{\pi_T,p_T}(s_t)|\, da_t\, ds_t \\ &= \mathbb{E}_{s_t \sim d_t^{\pi_S,p_S}}\left[ D_{TV}(\pi_S(\cdot|s_t, z_{S,t}), \pi_T(\cdot|s_t, z_{T,t})) \right] + D_{TV}(d_t^{\pi_S,p_S}, d_t^{\pi_T,p_T}) \end{aligned} \tag{A.20}$$

Substitute this bound into Eq. (A.19), we arrive at the desired result. $\qquad\square$

## A.4. Trajectory Divergence Lemmas

Implicit methods use an encoder to map trajectory segments to latent vectors. To quantify the Lipschitz continuity for encoder, we first establish the following lemma on divergence between two trajectories, and further extend it to divergence between two segments from different trajectories with the same initial state.

**Lemma A.3** (Trajectory Divergence Lemma). *Let $p(\cdot|s,a)$, $\pi(\cdot|s,z)$ denote the transition probability and adaptive policy in a domain respectively. Let $p(\tau)$ and $\tilde{p}(\tau)$ be the probability distributions over trajectories of length $H$ induced by $(\pi, p)$ and $(\tilde{\pi}, \tilde{p})$ respectively. The Total Variation distance between these distributions is bounded by:*

$$D_{TV}(p(\tau), \tilde{p}(\tau)) \leq \sum_{t=0}^{H-1} \mathbb{E}_{s_t \sim \tilde{d}_t} \left[ D_{TV}(\pi(\cdot|s_t, z_t), \tilde{\pi}(\cdot|s_t, \tilde{z}_t)) + \mathbb{E}_{a_t \sim \tilde{\pi}_t} \left[ D_{TV}(p(\cdot|s_t, a_t), \tilde{p}(\cdot|s_t, a_t)) \right] \right] \tag{A.21}$$

*where $\tilde{d}_t$ is the state marginal distribution induced by following $(\tilde{\pi}, \tilde{p})$ for $t$ steps, $z_t$ and $\tilde{z}_t$ are latent codes inferred from trajectory windows produced by $(\pi, p)$ and $(\tilde{\pi}, \tilde{p})$ respectively.*

*Proof.* We construct a sequence of interpolating distributions $\{p^{(k)}\}_{k=0}^{H}$, such that $p^{(0)}(\tau) = p(\tau)$ (Source) and $p^{(H)}(\tau) = \tilde{p}(\tau)$ (Target). For $1 \leq k \leq H - 1$, $p^{(k)}(\tau)$, $p^{(k)}$ is generated by following $(\tilde{\pi}, \tilde{p})$ for the first $k$ steps, and $(\pi, p)$ for the remaining $H - k$ steps:

$$p^{(k)}(\tau) = p(s_0) \left( \prod_{t=0}^{k-1} \tilde{\pi}(a_t|s_t, \tilde{z}_t) \tilde{p}(s_{t+1}|s_t, a_t) \right) \left( \prod_{t=k}^{H-1} \pi(a_t|s_t, z_t) p(s_{t+1}|s_t, a_t) \right) \tag{A.22}$$

although actions from $\pi$ are not used for the first $k$ steps, we still query $\pi$ to obtain trajectory history $\{s_{\leq k}, a_{<k}\}$, which are used to produce latents $z_{\geq k}$. For notational convenience, denote $\pi(a_t|s_t, z_t)$ as $\pi_t$, and $p(s_{t+1}|s_t, a_t)$ as $p_t$. By the telescoping sum and triangle inequality:

$$D_{TV}(p(\tau), \tilde{p}(\tau)) = D_{TV}(p^{(0)}, p^{(H)}) \leq \sum_{k=0}^{H-1} D_{TV}(p^{(k)}, p^{(k+1)}) \tag{A.23}$$

The term $D_{TV}(p^{(k)}, p^{(k+1)})$ can be further expanded as:

$$D_{TV}(p^{(k)}, p^{(k+1)}) = \frac{1}{2} \int \left| (\pi_k p_k - \tilde{\pi}_k \tilde{p}_k) p(s_0) \prod_{t=0}^{k-1} \tilde{\pi}_t \tilde{p}_t \prod_{t=k+1}^{H-1} \pi_t p_t \right| d\tau \tag{A.24}$$

Let $h_k = (s_0, a_0, \ldots, s_k)$ denote the history up to $s_k$ by following $(\tilde{\pi}, \tilde{p})$. The prefix density is $p(h_k) = p(s_0) \prod_{t=0}^{k-1} \tilde{\pi}_t \tilde{p}_t$ and the future conditional density is $p(\tau_{>k+1}|s_{k+1}) = \prod_{t=k+1}^{H-1} \pi_t p_t$. These two parts are shared for distributions $p^{(k)}$ and $p^{(k+1)}$. The difference arises solely at step $k$. Consider this difference term:

$$|\pi_k p_k - \tilde{\pi}_k \tilde{p}_k| \leq |\pi_k - \tilde{\pi}_k| p_k + |p_k - \tilde{p}_k| \tilde{\pi}_k \tag{A.25}$$

Integrate over the future variables $\tau_{>k+1}$ in Eq. (A.24), then plug in inequality (A.25), we arrive at:

$$D_{TV}(p^{(k)}, p^{(k+1)}) \leq \int p(h_k) \left[ \frac{1}{2} \iint \left( |\pi_k - \tilde{\pi}_k| p_k + |p_k - \tilde{p}_k| \tilde{\pi}_k \right) da_k ds_{k+1} \right] dh_k \tag{A.26}$$

The first term evaluates to:

$$\frac{1}{2} \iint |\pi_k - \tilde{\pi}_k| p_k \, da_k ds_{k+1} = \frac{1}{2} \int |\pi_k - \tilde{\pi}_k| \underbrace{\left( \int p_k \, ds_{k+1} \right)}_{1} da_k = D_{TV}(\pi_k, \tilde{\pi}_k) \tag{A.27}$$

The second term evaluates to:

$$\frac{1}{2} \iint |p_k - \tilde{p}_k| \tilde{\pi}_k \, da_k ds_{k+1} = \int \tilde{\pi}_k \underbrace{\left( \frac{1}{2} \int |p_k - \tilde{p}_k| \, ds_{k+1} \right)}_{D_{TV}(p_k, \tilde{p}_k)} da_k = \mathbb{E}_{a_k \sim \tilde{\pi}_k} [D_{TV}(p_k, \tilde{p}_k)] \tag{A.28}$$

Substituting these back into Eq. (A.26), the integral over $h_k$ becomes the expectation over the state distribution induced by the target parameters up to step $k$, denoted $\mathbb{E}_{s_k \sim \tilde{d}_k}$. Summing over $k$ we obtain the final bound. $\qquad \square$

**Lemma A.4** (Window Distribution Divergence Lemma)**.** *Let $\rho_t(\pi, p)$ and $\rho_t(\tilde{\pi}, \tilde{p})$ be the marginal distributions of trajectory windows starting at state $s_{t-H}$ with length $H$ (defined in Eq. (A.3)). The Total Variation divergence between these window distributions is bounded by:*

$$D_{TV}(\rho_t(\pi, p), \rho_t(\tilde{\pi}, \tilde{p})) \le D_{TV}(d_{t-H}^{\pi, p}(s_{t-H}), d_{t-H}^{\tilde{\pi}, \tilde{p}}(s_{t-H}))$$

$$+ \sum_{k=t-H}^{t-1} \mathbb{E}_{s_k \sim \tilde{d}_k}\left[D_{TV}(\pi(\cdot|s_k, z_k), \tilde{\pi}(\cdot|s_k, \tilde{z}_k)) + \mathbb{E}_{a_t \sim \tilde{\pi}_k}\left[D_{TV}(p(\cdot|s_k, a_k), \tilde{p}(\cdot|s_k, a_k))\right]\right]$$

(A.29)

*where $\tilde{d}_k$ is the state marginal distribution induced by following $(\tilde{\pi}, \tilde{p})$ for $k$ steps, $z_k$ and $\tilde{z}_k$ are latent codes inferred from trajectory windows produced by $(\pi, p)$ and $(\tilde{\pi}, \tilde{p})$ respectively.*

*Proof.* The difference between trajectory window and full trajectory is that different trajectories have the same initial distribution, but different windows start at different initial state, whose distribution is given by $d_{t-H}^{\pi, p}(s_{t-H})$ as specified in Eq. (A.3). For notational convenience, we shift the time index by $t - H$ and denote $s_{t-H}$ as $s_0$. Extending the proof of Trajectory Difference Lemma (Lemma A.3), for window $w_t$ starting at state $s_{t-H}$ with length $H$, we construct a sequence of interpolating distributions $\{p^{(k)}\}_{k=-1}^{H}$, satisfying:

$$p^{(-1)}(w) = d_0^{\pi, p}(s_0)\left(\prod_{i=0}^{H-1} \pi(a_i|s_i, z_i)p(s_{i+1}|s_i, a_i)\right)$$

(A.30)

$$p^{(0)}(w) = d_0^{\tilde{\pi}, \tilde{p}}(s_0)\left(\prod_{i=0}^{H-1} \pi(a_i|s_i, z_i)p(s_{i+1}|s_i, a_i)\right)$$

(A.31)

$$p^{(k)}(w) = d_0^{\tilde{\pi}, \tilde{p}}(s_0)\left(\prod_{i=0}^{k-1} \tilde{\pi}(a_i|s_i, \tilde{z}_i)\tilde{p}(s_{i+1}|s_i, a_i)\right)\left(\prod_{i=k}^{H-1} \pi(a_i|s_i, z_i)p(s_{i+1}|s_i, a_i)\right)$$

(A.32)

Therefore, $p^{(H)}(w)$ is the marginal window distribution in target main, and $p^{(-1)}(w)$ is the marginal in source domain. Again, $z_i$ and $\tilde{z}_i$ are latent codes inferred from trajectory windows produced by $(\pi, p)$ and $(\tilde{\pi}, \tilde{p})$ respectively. By the telescoping sum and triangle inequality:

$$D_{TV}(\rho_S, \rho_T) = D_{TV}(p^{(-1)}, p^{(H)}) \le \underbrace{D_{TV}(p^{(-1)}, p^{(0)})}_{\text{Initialization Shift}} + \underbrace{\sum_{k=0}^{H-1} D_{TV}(p^{(k)}, p^{(k+1)})}_{\text{Window Transition Shift}}$$

(A.33)

where Initialization Shift calculates to:

$$D_{TV}(p^{(-1)}, p^{(0)}) = D_{TV}(d^{\pi, p}(s_0), d^{\tilde{\pi}, \tilde{p}}(s_0))$$

(A.34)

Further using Lemma A.3 to bound Window Transition Shift, and shift time index by $t - H$, we obtain the final result. □

**Assumption A.3.** There exists a constant $C_d < 1$ such that $\sup_t D_{TV}(d_t^{\pi_S, p_S}(s_t), d_t^{\pi_T, p_T}(s_t)) \le C_d$.

**Lemma A.5** (Window Divergence Bound)**.** *Denote $\Delta_t^{\rho} := D_{TV}(\rho_t(\pi, p), \rho_t(\tilde{\pi}, \tilde{p}))$. Let $\mathcal{M}_S$ and $\mathcal{M}_T$ be two MDPs that are $\epsilon_p$-close in dynamics. If adaptive policy $\pi$ is $L_\pi$-smooth, encoder $E$ is $L_E$-smooth, $\delta_e$-consistent in both MDPs, then under Assumption A.3 and $L_\pi L_E H \neq 1$, the following bound holds:*

$$\Delta_t^{\rho} \le \frac{b}{1 - aH} + Mr_0^t$$

(A.35)

*where $a := L_\pi L_E$, $b := C_d + H(2L_\pi \delta_e + \epsilon_p)$ and $M := \max_{0 \le k < H}(\Delta_k^{\rho} - b/(1 - aH))/r_0^k$ are all constants. $r_0 \in \mathbb{R}^+$ is the unique root of $r^H(r - 1) = a(r^H - 1)$.*

*Proof.* For notational convenience, denote $D_{TV}(\rho_t(\pi, p), \rho_t(\tilde{\pi}, \tilde{p}))$ as $\Delta_t^\rho$, $L_\pi L_E$ as $a$ and $C_d + H(2L_\pi \delta_e + \epsilon_p)$ as $b$. Assume $\Delta_k^\rho = 0$ when $k \le 0$. We first relate policy divergence to window divergence:

$$
\begin{aligned}
D_{TV}(\pi(\cdot|s_t, z_t), \tilde{\pi}(\cdot|s_t, \tilde{z}_t)) &\overset{(1)}{\le} L_\pi \|z_t - \tilde{z}_t\|_2 \\
&\overset{(2)}{\le} L_\pi [\|z_t - \mu_t(\pi, p)\|_2 + \|\mu_t(\pi, p) - \mu_t(\tilde{\pi}, \tilde{p})\|_2 + \|\mu_t(\tilde{\pi}, \tilde{p}) - \tilde{z}_t\|_2] \\
&\overset{(3)}{\le} L_\pi (2\delta_e + L_E \Delta_t^\rho)
\end{aligned}
\tag{A.36}
$$

where inequality (1) follows $L_\pi$-smoothness of policy, inequality (2) relates two latent codes to their corresponding latent centroids, and inequality (3) is derived from the definition of $\delta_e$-consistency and $L_E$-smoothness of encoder. With this bound and Assumption A.3, Lemma A.4 evaluates to:

$$
\Delta_t^\rho \le C_d + \sum_{k=t-H}^{t-1} L_\pi(2\delta_e + L_E \Delta_k^\rho) + \epsilon_p \Leftrightarrow \Delta_t^\rho \le a \sum_{k=t-H}^{t-1} \Delta_k^\rho + b
\tag{A.37}
$$

Let $c_k = \Delta_k^\rho - \frac{b}{1-aH}$, the recurrence relation for $c_k$ is:

$$
c_t \le a \sum_{k=t-H}^{t-1} c_k, \quad \forall t \ge H
\tag{A.38}
$$

Consider the equality case $c_t = a \sum_{k=t-H}^{t-1} c_k$, the characteristic equation is:

$$
g(r) := \frac{r^H}{1 + r + \ldots + r^{H-1}} = a
\tag{A.39}
$$

Function $g(r)$ is strictly increasing on $(0, \infty)$. Moreover, $\lim_{r \to 0^+} g(r) = 0$, $g(1) = \frac{1}{H}$, $\lim_{r \to \infty} g(r) = \infty$. Hence for any $a > 0$ there exists a unique $r_0 > 0$ such that $g(r_0) = a$. In particular, if $aH > 1$ then $r_0 > 1$; if $aH = 1$ then $r_0 = 1$; if $aH < 1$ then $r_0 < 1$. Let $M = \max_{0 \le k < H}(c_k/r_0^k)$, we prove by induction that $c_t \le Mr_0^t$ for all $t$. For base case $t < H$, the inequality holds by the definition of $M$. For inductive step, assume $c_k \le Mr_0^k$ for all $k < t$ with $t \ge H$. Then:

$$
c_t \le a \sum_{k=t-H}^{t-1} Mr_0^k = Mar_0^{t-H}(1 + r_0 + \ldots + r_0^{H-1}) = Mr_0^t
\tag{A.40}
$$

which completes the inductive step and therefore concludes the proof. $\square$

*Remark* 2. Several explanations regarding this bound: (1) Upper-bound sequence $f(t)$ converges only when $aH < 1$, the resulting value is $\lim_{t \to \infty} f(t) = \frac{b}{1-aH}$; (2) Given $r_0^H(r_0 - 1) = a(r_0^H - 1)$, we calculate the rate of change w.r.t. $L_E$:

$$
\frac{\partial r_0}{\partial L_E} = L_\pi \frac{(r_0^H - 1)^2}{r_0^{H-1}[r_0^{H+1} - (H+1)r_0 + H]}
\tag{A.41}
$$

which is strictly positive for $r_0 > 0$, $r_0 \ne 1$. Therefore $r_0$ is an increasing function of $L_E$, decreasing $L_E$ reduces $r_0$ and leads to a tighter bound; and (3) Since $D_{TV}(\cdot, \cdot) \le 1$ by definition, and when $aH > 1$, this exponential upper-bound quickly exceeds 1, making it seemingly useless. Notice the goal here is not to provide a tight bound for $\Delta_t^\rho$, but to reveal the dominating factor for window divergence (encoder Lipschitz constant $L_E$) that guides algorithmic design.

### A.5. Target Domain Regret Bound

**Assumption A.4.** The latent-conditioned policy $\pi_S = \pi(\cdot|s, z_S)$ approximates the source oracle $\pi_S^* = \pi(\cdot|s)$ within a bounded error $\delta_{train}$ during training.

**Theorem A.1** (Target Domain Regret Bound). *Let $\mathcal{M}_S$ and $\mathcal{M}_T$ be two MDPs that are $\epsilon_p$-close in dynamics. If policy $\pi$ is $L_\pi$-smooth (Definition A.3), encoder $E$ is $L_E$-smooth (Definition A.6), $\delta_e$-consistent (Definition A.5) in both MDPs, Assumption A.3 and A.4 hold, then the bound for target domain regret is monotonically increasing with respect to $\delta_e$ and $L_E$.*

*Proof.* By the triangle inequality, performance difference between adaptive policy and oracle can be decomposed into:

$$\left\|V_T^{\pi_T^*} - V_T^{\pi_T}\right\|_\infty \leq \underbrace{\left\|V_T^{\pi_T^*} - V_S^{\pi_S^*}\right\|_\infty}_{\text{(I) Oracle Difference}} + \underbrace{\left\|V_S^{\pi_S^*} - V_S^{\pi_S}\right\|_\infty}_{\text{(II) Training Error}} + \underbrace{\|V_S^{\pi_S} - V_T^{\pi_T}\|_\infty}_{\text{(III) Policy Adaptation}} \tag{A.42}$$

where $\pi_S^*$ denotes the oracle trained in source, and $\pi_S$ refers to the adaptive policy deployed in source domain. In Eq. (A.42), term (II) is bounded by Assumption A.4. For term (I), notice:

$$V_T^{\pi_T^*} - V_S^{\pi_S^*} = \left(V_T^{\pi_T^*} - V_S^{\pi_T^*}\right) + \left(V_S^{\pi_T^*} - V_S^{\pi_S^*}\right) \leq V_T^{\pi_T^*} - V_S^{\pi_T^*} \tag{A.43}$$

where the inequality stems from $V_S^{\pi_T^*}(s) - V_S^{\pi_S^*}(s) \leq 0$ by the definition of source oracle. Therefore term (I) can be bounded by standard Simulation Lemma (Lemma A.1):

$$\text{Term (I)} \leq \frac{2\gamma R_{max}}{(1-\gamma)^2}\epsilon_p \tag{A.44}$$

For term (III), we resort to Extended Performance Difference Lemma (Lemma A.2), combined with Assumption A.3:

$$\|V_T^{\pi_T} - V_S^{\pi_S}\|_\infty \leq 2R_{max} \sum_{t=0}^\infty \gamma^t \left(C_d + \sup_{s_t} \mathbb{E}_{s_t \sim d_t^{\pi_S,p_S}} \left[D_{TV}(\pi_S(\cdot|s_t, z_{S,t}), \pi_T(\cdot|s_t, z_{T,t}))\right]\right)$$

$$\leq 2R_{max} \sum_{t=0}^\infty \gamma^t \left(C_d + 2L_\pi\delta_e + L_\pi L_E \Delta_t^\rho\right) \tag{A.45}$$

$$= 2R_{max} \left(\frac{C_d + 2L_\pi\delta_e}{1-\gamma} + L_\pi L_E \sum_{t=0}^\infty \gamma^t \Delta_t^\rho\right)$$

where the second inequality follows inequality (A.36). Further using Lemma A.5 to bound $\Delta_t^\rho$, we obtain:

$$\|V_T^{\pi_T} - V_S^{\pi_S}\|_\infty \leq 2R_{max} \left(\frac{C_d + 2L_\pi\delta_e}{1-\gamma} + \frac{ab}{(1-\gamma)(1-aH)} + aM\sum_{t=0}^\infty (\gamma r_0)^t\right) \tag{A.46}$$

where $a = L_\pi L_E$, $b = C_d + H(2L_\pi\delta_e + \epsilon_p)$, $M = \max_{0 \leq k < H}(\Delta_k^\rho - b/(1-aH))/r_0^k$, and $r_0 \in \mathbb{R}^+$ is the unique root of $r^H(r-1) = a(r^H - 1)$. Combining these bounds we get the closed-form expression for target domain regret:

$$\left\|V_T^{\pi_T^*} - V_T^{\pi_T}\right\|_\infty \leq \frac{2\gamma R_{max}}{(1-\gamma)^2}\epsilon_p + \delta_{train} + 2R_{max}\left(\frac{C_d + 2L_\pi\delta_e}{1-\gamma} + \frac{ab}{(1-\gamma)(1-aH)} + aM\sum_{t=0}^\infty (\gamma r_0)^t\right) \tag{A.47}$$

To analyze how this bound changes w.r.t. $\delta_e$ and $L_E$, we isolate the impact of $\|V_T^{\pi_T} - V_S^{\pi_S}\|_\infty$. Define the worst-case deterministic sequence $f(t)$ that bounds the actual divergence $\Delta_t^\rho$ in Eq. (A.45) and denote the resulting upper-bound as $F$:

$$F = 2R_{max}\left(\frac{C_d + 2L_\pi\delta_e}{1-\gamma} + L_\pi L_E \sum_{t=0}^\infty \gamma^t f(t)\right) \tag{A.48}$$

where $f(t)$ is generated by replacing the inequality in Eq. (A.37) with equality:

$$f(t) = a\sum_{k=t-H}^{t-1} f(k) + b \tag{A.49}$$

It is straightforward to prove via induction that $\partial f(t)/\partial a > 0$ and $\partial f(t)/\partial b > 0$, given the following relation:

$$\frac{\partial f(t)}{\partial a} = \sum_{k=t-H}^{t-1} f(k) + a\sum_{k=t-H}^{t-1} \frac{\partial f(k)}{\partial a}, \quad \frac{\partial f(t)}{\partial b} = a\sum_{k=t-H}^{t-1} \frac{\partial f(k)}{\partial b} + 1 \tag{A.50}$$

which further indicates:

$$\frac{\partial f(t)}{\partial \delta_e} = 2HL_\pi \frac{\partial f(t)}{\partial b} > 0, \quad \frac{\partial f(t)}{\partial L_E} = L_\pi \frac{\partial f(t)}{\partial a} > 0 \tag{A.51}$$

Therefore, consider the change rate of final upper-bound $F$:

$$\frac{\partial F}{\partial \delta_e} = 2R_{max}\left(\frac{2L_\pi}{1-\gamma} + L_\pi L_E \sum_{t=0}^{\infty} \gamma^t \frac{\partial f(t)}{\partial \delta_e}\right) > 0 \tag{A.52}$$

$$\frac{\partial F}{\partial L_E} = 2R_{max}L_\pi \sum_{t=0}^{\infty} \gamma^t \left(f(t) + L_E \frac{\partial f(t)}{\partial L_E}\right) > 0 \tag{A.53}$$

this completes the proof that target domain regret bound is a monotonically increasing function w.r.t. $\delta_e$ and $L_E$. $\qquad\square$

## A.6. InfoNCE Minimization of Encoder Lipschitz Constant

We will be using the following decomposition (Wang & Isola, 2020) during our analysis of InfoNCE Loss:

$$\lim_{N\to\infty} \mathcal{L}_{InfoNCE} \propto \underbrace{\mathbb{E}_{(w,w^+)\sim\rho_{pos}}\left[\|E(w) - E(w^+)\|_2^2\right]}_{\mathcal{L}_{align}} + \underbrace{\mathbb{E}_{(w,w^-)\sim\rho_{data}}\left[e^{-\|E(w)-E(w^-)\|_2^2}\right]}_{\mathcal{L}_{uniform}} \tag{A.54}$$

**Theorem A.2.** *Assume the measure of the support intersection $S = \mathrm{supp}(\pi, p) \cap \mathrm{supp}(\tilde{\pi}, \tilde{p})$ satisfies $\mathbb{P}(S) > \alpha$ for some $\alpha > 0$, then $L_E$ is strictly upper-bounded by the square root of $\mathcal{L}_{align}$.*

*Proof.* Distribution for positive samples can be explicitly written as $\rho_{pos} = \mathbb{E}_p\mathbb{E}_{\pi,t}\left[\rho_t(\pi, p)\right]$, namely the "assemble" of all window distributions produced under the same dynamics. Since positive samples only condition on dynamics and not time steps, we define a "global latent centroid" that is time-invariant (as opposed to prior time-indexed latent centroid):

$$\bar{\mu}(\pi, p) = \mathbb{E}_t\left[\mu_t(\pi, p)\right] \tag{A.55}$$

and further define the Alignment Loss under a fixed policy $\pi$ and dynamics $p$ (the corresponding positive samples distribution is $\rho_{pos}(\pi, p) = \mathbb{E}_t[\rho_t(\pi, p)]$):

$$\mathcal{L}_{align}(\pi, p) = \mathbb{E}_{(w,w^+)\sim\rho_{pos}(\pi,p)}\left[\left\|E(w) - E(w^+)\right\|_2^2\right] \tag{A.56}$$

The proof proceeds by relating the Alignment Loss to the radius of the probability mass concentration (Intra-Cluster Tightness) and then demonstrating that overlapping dynamics must share a latent region (Inter-Cluster Closeness).

### 1. Intra-Cluster Tightness
Utilizing the variance identity $\mathbb{E}[\|X - Y\|^2] = 2\,\mathrm{Var}(X)$ for i.i.d. variables sampled from $\rho_{pos}(\pi, p)$ (we abbreviate it to $\rho_{\pi,p}$ to shorten notation), we have:

$$\mathbb{E}_{w\sim\rho_{\pi,p}}\left[\|E(w) - \bar{\mu}(\pi, p)\|_2^2\right] = \frac{1}{2}\mathbb{E}_{(w,w^+)\sim\rho_{\pi,p}}\left[\|E(w) - E(w^+)\|_2^2\right] = \frac{1}{2}\mathcal{L}_{align}(\pi, p) \tag{A.57}$$

Define the cluster variance $\sigma_{\pi,p}^2 := \mathcal{L}_{align}(\pi, p)/2$. According to multivariate Chebyshev's inequality, for a random trajectory window $w \sim \rho_{\pi,p}$, the probability that its latent embedding lies outside a radius $k\sigma_{\pi,p}$ is bounded:

$$\mathbb{P}_{w\sim\rho_{\pi,p}}\left(\|E(w) - \bar{\mu}(\pi, p)\|_2 \geq k\sigma_{\rho_{\pi,p}}\right) \leq \frac{d}{k^2} \tag{A.58}$$

where $d$ is the latent dimension. Under the confidence threshold:

$$\epsilon_{align}(\pi, p) := \sqrt{\frac{d \cdot \mathcal{L}_{align}(\pi, p)}{\alpha}} \tag{A.59}$$

then with high probability ($> 1 - \alpha/2$), any sample $w$ from $\rho_{\pi,p}$ satisfies $\|E(w) - \bar{\mu}(\pi, p)\|_2 \leq \epsilon_{align}(\pi, p)$.

**2. Inter-Cluster Closeness**

Consider the physical support intersection $S = \text{supp}(\pi, p) \cap \text{supp}(\tilde{\pi}, \tilde{p})$. We prove that there exists a "bridge sample" $w^* \in S$ that lies within the high-density regions of both clusters. Let $H_{\pi,p}$ be the set of trajectory windows where $\|E(w) - \bar{\mu}(\pi, p)\| \leq \epsilon_{align}(\pi, p)$, and $H_{\pi,p}^c$ be its complement set. Let $H_{\tilde{\pi},\tilde{p}}$ and $H_{\tilde{\pi},\tilde{p}}^c$ be the equivalent sets for dynamics and policy pair $(\tilde{\pi}, \tilde{p})$. From Step 1, $\mathbb{P}(H_{\pi,p}^c) \leq \alpha/2$ and $\mathbb{P}(H_{\tilde{\pi},\tilde{p}}^c) \leq \alpha/2$. By the union bound, the measure of "outliers" is at most $\alpha$. Since the intersection $S$ has measure greater than $\alpha$, the set of valid bridges $B = S \cap H_{\pi,p} \cap H_{\tilde{\pi},\tilde{p}} \neq \emptyset$. Therefore, there exists at least one trajectory window $w^*$ such that:

$$\|E(w^*) - \bar{\mu}(\pi, p)\|_2 \leq \epsilon_{align}(\pi, p) \quad \wedge \quad \|E(w^*) - \bar{\mu}(\tilde{\pi}, \tilde{p})\|_2 \leq \epsilon_{align}(\tilde{\pi}, \tilde{p}) \tag{A.60}$$

Using $w^*$ as the anchor in the triangle inequality:

$$\|\bar{\mu}(\pi, p) - \bar{\mu}(\tilde{\pi}, \tilde{p})\|_2 \leq \|\bar{\mu}(\pi, p) - E(w^*)\|_2 + \|E(w^*) - \bar{\mu}(\tilde{\pi}, \tilde{p})\|_2 \leq \epsilon_{align}(\pi, p) + \epsilon_{align}(\tilde{\pi}, \tilde{p}) \tag{A.61}$$

Since the encoder Lipschitz constant is defined in terms of time-dependent latent centroids, we further bound $\bar{\mu} - \mu_t$ via Jensen's inequality (utilizing the convexity of $L_2$-norm)

$$\|\bar{\mu}(\pi, p) - \mu_t(\pi, p)\|_2 = \left\|\mathbb{E}_{w_t \sim \rho_t(\pi,p)}\left[E(w_t) - \bar{\mu}(\pi, p)\right]\right\|_2 \leq \mathbb{E}_{w_t \sim \rho_t(\pi,p)}\left[\|E(w_t) - \bar{\mu}(\pi, p)\|_2\right] \tag{A.62}$$

Utilizing the property that $\mathbb{E}[X] \leq \sqrt{\mathbb{E}[X^2]}$, we obtain:

$$\mathbb{E}_{w_t \sim \rho_t(\pi,p)}\left[\|E(w_t) - \bar{\mu}(\pi, p)\|_2\right] \leq \sqrt{\mathbb{E}_{w_t \sim \rho_t(\pi,p)}\left[\|E(w_t) - \bar{\mu}(\pi, p)\|_2^2\right]} = \sqrt{\mathcal{L}_{align}(\pi, p)/2} \tag{A.63}$$

Therefore we bound the distance between time-indexed latent centroids:

$$\begin{aligned}\|\mu_t(\pi, p) - \mu_t(\tilde{\pi}, \tilde{p})\|_2 &\leq \|\bar{\mu}(\pi, p) - \mu_t(\pi, p)\|_2 + \|\bar{\mu}(\tilde{\pi}, \tilde{p}) - \mu_t(\tilde{\pi}, \tilde{p})\|_2 + \|\bar{\mu}(\pi, p) - \bar{\mu}(\tilde{\pi}, \tilde{p})\|_2 \\ &\leq 2\sqrt{\mathcal{L}_{align}}\left(\sqrt{1/2} + \sqrt{d/\alpha}\right)\end{aligned} \tag{A.64}$$

where the first inequality uses the triangle inequality, and the second inequality uses the fact that local Alignment Loss is smaller than global Alignment Loss. Substituting this bound into the definition of $L_E$:

$$L_E = \sup_t \sup_{(\pi,p),(\tilde{\pi},\tilde{p})} \frac{\|\mu_t(\pi, p) - \mu_t(\tilde{\pi}, \tilde{p})\|_2}{D_{TV}(\rho_t(\pi, p), \rho_t(\tilde{\pi}, \tilde{p}))} \leq \frac{2\sqrt{\mathcal{L}_{align}}\left(\sqrt{1/2} + \sqrt{d/\alpha}\right)}{D_{TV}(\rho_t(\pi, p), \rho_t(\tilde{\pi}, \tilde{p}))} \tag{A.65}$$

Namely, given fixed window divergence during optimization, $L_E$ is strictly upper-bounded by $\sqrt{\mathcal{L}_{align}}$. $\qquad\square$

**Theorem A.3.** *Let the space of trajectory windows $\mathcal{W}$ be locally factorizable into dynamics-relevant features and nuisance features, such that trajectory window $w \approx (\mu, s)$. Then minimizing the InfoNCE Loss (Eq. (A.54)) implies minimizing the Frobenius norm of $\partial E/\partial s$.*

*Proof.* In our setting, a positive pair $(w, w^+)$ shares dynamics $\mu$ but differs in nuisance factors $s$ (e.g., initial state, noise). Define the perturbation $\delta = w^+ - w \approx (0, \Delta s)$. Using the first-order Taylor expansion of the encoder:

$$E(w^+) - E(w) \approx \mathbf{J}_E \delta \approx \frac{\partial E}{\partial \mu} \cdot 0 + \frac{\partial E}{\partial s} \cdot \Delta s \tag{A.66}$$

The Alignment Loss in Eq. (A.54) explicitly minimizes this difference, and since $\Delta s$ represents natural variations in the environment (which are non-zero), the optimization must decrease the partial derivative with respect to nuisance factors:

$$\min \mathcal{L}_{align} \implies \min \mathbb{E}\left[\left\|\frac{\partial E}{\partial s} \Delta s\right\|^2\right] \implies \min \left\|\frac{\partial E}{\partial s}\right\|_F^2 \tag{A.67}$$

Therefore, minimizing $\mathcal{L}_{InfoNCE}$ implies minimizing the Frobenius norm of $\partial E/\partial s$. $\qquad\square$

---

**Algorithm 1** Joint Latent Dynamics Representation and Policy Learning

1: **Input:** Buffer $\mathcal{D}$, Encoder $q_\phi$, Decoder $p_\theta$, Policy $\pi_\psi$, Batch size $N$, Weights $\lambda_1, \lambda_2, \beta$
2: **while** not converged **do**
3:    Collect transitions using $\pi_\psi$ and store in $\mathcal{D}$
4:    Sample batch $\mathcal{B} = \{(w_i, s_i, a_i, s'_i, r_i)\}_{i=1}^N$
5:    Infer latent distribution: $z_i \sim q_\phi(\cdot|w_i)$
6:    $\mathcal{L}_{\text{rec}} \leftarrow -\frac{1}{N}\sum_i \log p_\theta(s'_i \mid s_i, a_i, z_i)$
7:    $\mathcal{L}_{\text{KL}} \leftarrow \frac{1}{N}\sum_i D_{KL}(q_\phi(\cdot|w_i)\|\mathcal{N}(0, \mathbf{I}))$
8:    Combine VAE objective: $\mathcal{L}_{\text{VAE}} \leftarrow \mathcal{L}_{\text{rec}} + \beta\mathcal{L}_{\text{KL}}$
9:    Use $w_i$ under same dynamics to compute $\mathcal{L}_{\text{contrast}}$ with Eq. (12)
10:    # Assume latent is the same for $s_i$ and $s'_i$
11:    Augment observations: $x_i \leftarrow [s_i, z_i]$, $x'_i \leftarrow [s'_i, z_i]$
12:    Compute RL Loss $\mathcal{L}_{\text{rl}}(x_i, x'_i, a_i, r_i)$
13:    $\mathcal{L} \leftarrow \mathcal{L}_{\text{rl}} + \lambda_1\mathcal{L}_{\text{VAE}} + \lambda_2\mathcal{L}_{\text{contrast}}$
14:    Update $\phi, \theta, \psi$ via gradient descent on $\mathcal{L}$
15: **end while**

---

*Remark* 3. We can also prove the upper-bounding of $L_E$ from this gradient orthogonality perspective. Encoder Jacobian decomposes into orthogonal components corresponding to dynamics and nuisance features:

$$\|\mathbf{J}_E\|_F^2 = \left\|\frac{\partial E}{\partial \mu}\right\|_F^2 + \left\|\frac{\partial E}{\partial s}\right\|_F^2 \tag{A.68}$$

Since the Lipschitz constant $L_E$ is upper-bounded by the spectral norm (therefore also upper-bounded by the Frobenius norm) of the full Jacobian $\mathbf{J}_E$, applying InfoNCE leads to a tighter Lipschitz constant by strictly decreasing the total norm $\|\mathbf{J}_E\|$, via the minimization of $\|\partial E/\partial s\|_F$.

## B. Implementation Details

### B.1. Training Procedure

We provide the overall algorithmic process in Algorithm 1.

### B.2. Dynamics Randomization

*Table B.1.* Statistics of Environment Dimensions and Randomized Dynamics Parameters

| | Dimensions | | Rand. Params. Count | | | | |
|---|---|---|---|---|---|---|---|
| Environment | Obs. | Act. | Fric. | Mass | Damp. | Torq. | Total |
| Hopper | 11 | 3 | — | 4 | 3 | 1 | 8 |
| Walker2d | 17 | 6 | — | 7 | 6 | 1 | 14 |
| HalfCheetah | 17 | 6 | 9 | 7 | 6 | 1 | 23 |
| Ant | 27 | 8 | 14 | 13 | 8 | 1 | 36 |

*Table B.2.* Dynamics Randomization Ranges

| Parameter | Hopper | Walker2d | HalfCheetah | Ant |
|---|---|---|---|---|
| Mass | $[0.5, 2.0)$ | $[0.5, 2.0)$ | $[0.5, 2.0)$ | $[0.5, 2.0)$ |
| Damp. | $[0.5, 2.0)$ | $[0.5, 2.0)$ | $[0.5, 2.0)$ | $[0.5, 2.0)$ |
| Torq. | $[0.5, 1.5)$ | $[0.5, 1.5)$ | $[0.5, 1.5)$ | $[0.5, 1.5)$ |
| Fric. | — | — | $[0.4, 1.0)$ | $[0.2, 1.0)$ |

To assess robustness against OOD dynamics, we introduce randomization across four core physical properties: body mass, joint damping, slide friction, and torque scale. Among these, body mass affects the inertial resistance and momentum of the robot links; joint damping alters the internal friction and energy dissipation within joints; slide friction changes the traction between the robot and the ground, influencing acceleration and stability; and torque scale simulates actuator strength or weakness, affecting the control authority. Environment statistics, including observation dimension, control dimension and number of dynamics parameters randomized are collected in Table B.1. Aliases used: **Obs.** for observation dimension, **Act.** for action dimension, **Damp.** for joint damping, **Fric.** for slide friction, **Torq.** for torque scale, **Total** for the total number of dynamics parameters randomized. The scaling factors are sampled uniformly from the ranges specified in Table B.2. The sampling ranges are calibrated to induce significant trajectory diversity while maintaining task feasibility.

### B.3. Network Architecture

We implement the policy, critic, and auxiliary dynamics networks using standard multi-layer perceptrons (MLPs) and convolutional encoders.

**Dynamics Encoder ($q_\phi$):** The encoder processes the trajectory history window $w_t$ using a Temporal Convolutional Network (TCN (Bai et al., 2018)). It consists of three convolutional layers with kernel sizes $[8, 5, 5]$ and strides $[4, 1, 1]$, followed by a linear projection to the latent mean $\mu_\phi$ and log-standard deviation $\sigma_\phi$. We use ReLU activations for the intermediate layers.

**Dynamics Decoder ($p_\theta$):** The decoder is a 3-layer MLP with hidden units $[256, 256]$ (or $[512, 512]$ for Ant) and ReLU activations. It takes the latent $z_t$ and the current state-action pair $(s_t, a_t)$ to predict the next state residual $\Delta s = s_{t+1} - s_t$. We normalize the target $\Delta_s$ using running mean and std calculated dynamically during training, to make the decoder loss invariant to state element changing rate. Since the latent $z_t$ from encoder is theoretically unbounded, we apply Layer-Normalization (LN (Ba et al., 2016)) to latent vectors to improve training stability.

**Policy and Critic:** Both the actor $\pi_\psi$ and critic $Q_\omega$ utilize 2-layer MLPs with 256 hidden units. They receive the concatenation of the current state $s_t$ and the latent context $z_t$ as input (critic additionally receives $a_t$ as input). As with decoder, we apply LN to the input latent vectors.

### B.4. Contrastive Training Details

**Latent Dropout:** When conditioning a policy on a latent context vector extracted from a temporal history, the actor network can become overly reliant on the global latent representation, making the policy prone to failure when the latent context is not precise (e.g., in OOD cases). To increase the robustness of the policy, we implement latent dropout during both training and inference:

$$z_{rl} = z \odot M, \quad M \sim \text{Bernoulli}(1 - p) \tag{B.1}$$

where $p$ is the dropout probability. We empirically found that a conservative dropout rate of $p = 0.1$ effectively regularizes the policy across all environments without destroying the inferred dynamics context.

**Anchor Dynamics Strategy:** To stabilize the contrastive learning process in vectorized environments, we implement an "Anchor Dynamics" sampling strategy. When collecting experience with $N$ parallel environments, we ensure that a subset of environments are reset to a fixed "anchor" dynamics parameter ID at the beginning of data collection. This ensures that every training batch contains multiple trajectories generated from the exact same underlying physics. By anchoring the batch distribution, the InfoNCE objective shifts from "separating all random samples" to the easier task of "separating diverse clusters from a known anchor". We found this strategy significantly accelerates convergence of the geometric structure, as the encoder can quickly latch onto the consistent features of the anchor trajectories to form a central reference cluster. We prevent the anchor from being overwritten by newly incoming data once the buffer is full.

### B.5. Environment Wrappers and Modifications

**Ant Body Frame Wrapper:** For 3D omnidirectional agents like Ant, the standard observation space includes the global Cartesian velocity. However, from the perspective of underlying physical dynamics (e.g., damping or mass), moving forward at a given speed is functionally identical to moving backward or sideways at that same speed. If global velocities are passed directly to the trajectory encoder, this directional symmetry would cause the contrastive objective to split functionally equivalent dynamics into separate clusters based purely on heading. To resolve this, we wrap the environment to project the global velocity into the agent's local body frame, ensuring the latent representation is invariant to the agent's global orientation.

**Walker2d Torque Wrapper:** Same as (Yu et al., 2019), we observed that the default ankle torque limits for walker2d

$([-100, 100])$ allows the agent to exploit unrealistic "hopping" gaits. To facilitate learning of natural walking gait, we reduce ankle joint torque limits to $[-20, 20]$.

### B.6. Hyperparameter Choices

To balance the competing objectives of reward maximization, reconstruction, and contrastive shaping, we employ a weighted loss function:

$$\mathcal{L} = \lambda_{\text{rl}}\mathcal{L}_{\text{rl}} + \lambda_{\text{VAE}}\mathcal{L}_{\text{VAE}} + \lambda_{\text{contrast}}\mathcal{L}_{\text{contrast}} \tag{B.2}$$

We fix $\lambda_{\text{rl}} = 1.0$, $\lambda_{\text{contrast}} = 1.0$ and $\lambda_{\text{VAE}} = 5.0$ across all environments, while tuning the reward scale to ensure gradient magnitude consistency. We utilize a trajectory sequence length of $H = 50$ and a contrastive temperature of $\tau = 1.0$. Crucially, contrastive learning requires a sufficient diversity of positive and negative pairs within each update step. We therefore collect 2048 samples before running 2048 steps of gradient descent with a batch size of 256, ensuring the optimizer sees a dense sampling of the current policy's trajectory distribution. To prevent posterior collapse during VAE training, we adopt cost annealing (Bowman et al., 2016), with KL $\beta$ increasing linearly from 0 to the preset weight during the first $50\%$ of training, and remains constant afterwards. The networks are optimized using the Adam optimizer with a learning rate of $3 \times 10^{-4}$ for Hopper, Walker2d, HalfCheetah and $1 \times 10^{-4}$ for Ant. A summary of the environment-specific hyperparameters is provided in Table B.3.

*Table B.3.* Key environment-specific hyperparameters for LDG.

| Parameter | Hopper | Walker2d | HalfCheetah | Ant |
|---|---|---|---|---|
| Latent Dimension ($d_z$) | 3 | 3 | 3 | 5 |
| Reward Scale | 0.008 | 0.004 | 0.004 | 0.006 |
| KL Max $\beta$ (Annealing) | 0.1 | 0.07 | 0.07 | 0.04 |

Refer to Section C for a complete ablation of hyperparameter choices.

### B.7. Baseline Implementations

Our baseline RMA and SAC shares the exact policy and critic backbone as LDG. When training RMA, the environment extractor is a two-layer MLP with hidden dimension being $[256, 128]$. We use the same latent dimension for RMA and LDG. We also use Adam optimizer for RMA, and set learning rate to $3.0 \times 10^{-4}$ for Walker2d, HalfCheetah, Ant environments, and $1.0 \times 10^{-4}$ for Hopper environment.

For RMA adaptation phase, we construct the adaptation network to have the same architecture as LDG dynamics encoder $q_\phi$. Adaptation buffer size is set to $1.0 \times 10^5$, and a total of $4.0 \times 10^5$ transitions are collected during adaptation phase. We take one gradient step for every environment step with a batch size of 256, using Adam optimizer with learning rate $5.0 \times 10^{-4}$. Empirically, we found the adaptation network to steadily converge under this parameter setting.

For SO-CMA, we followed the original paper (Yu et al., 2019) and set rollout number to 1, population size to 7 and max iteration number to 6. For every candidate latent, the policy rollouts for 1 episode.

## C. Hyperparameter Tuning and Ablation Study

### C.1. Hyperparameter Ablations

Apart from the reward curves, we additionally evaluated the performance of different horizon $H$ and latent dimension $d_z$ configurations in ID, OOD, Var. Env and Struct. Fail settings.

**Latent Dimension ($d_z$):** Quantitative results are shown in Table C.1. Insufficient capacity (e.g., $d_z = 2$) catastrophically bottlenecks information, causing severe performance degradation across both environments. Conversely, excessive capacity ($d_z \geq 7$) introduces manifold sparsity that destabilizes the contrastive uniformity loss, resulting in brittle policies. While extreme over-parameterization ($d_z = 15$) allows the policy to partially recover through sheer parameter volume, it lacks the efficiency of a properly constrained bottleneck. Ultimately, $d_z = 3$ (for Walker2D with observation dimension 17) and $d_z = 5$ (for Ant with observation dimension 27) offer the optimal balance of representational expressivity and geometric tractability.

*Table C.1.* Ablation of latent dimension ($d_z$) across Walker2d and Ant environments. Rewards are averaged over 5 random seeds. Top-1 result is highlighted using bold numbers.

| Environment | $d_z$ | ID | OOD Damping | | Var. Env | Struct. Fail |
|---|---|---|---|---|---|---|
| | | | **0.3** | **2.2** | | |
| | 2 | 3845 | 3980 | 3970 | 3972 | 662 |
| | 3 (Ours) | **4884** | 5342 | 5349 | **5293** | 952 |
| **Walker2d** | 7 | 4254 | 4391 | 4397 | 4512 | 417 |
| | 10 | 4254 | 4848 | 4658 | 4680 | **1284** |
| | 15 | 4011 | **5513** | **5486** | 4886 | 369 |
| | 2 | 2097 | 3552 | 3545 | 2240 | **1468** |
| | 5 (Ours) | 5183 | 4797 | **4789** | **3462** | 1003 |
| **Ant** | 7 | 3253 | 1045 | 1957 | 582 | 579 |
| | 10 | 4481 | 1151 | 2669 | 672 | 522 |
| | 15 | **5247** | **3251** | 5343 | 3462 | 682 |

**Trajectory Horizon Length** ($H$): We also ablate the trajectory horizon length, as shown in Table C.2. The horizon dictates the temporal context window available for dynamics inference. Extremely short horizons ($H = 10$, corresponding to 0.08s of simulation time for Ant and 0.5s for Walker2d) lack sufficient temporal context for the encoder to accurately infer underlying physical parameters. In contrast, excessively long horizons ($H = 70$) integrate too much historical data, which dilutes the agent's immediate reactivity to recent state transitions. We find that $H = 50$ provides an optimal balance. Notably, while it might theoretically seem that shorter horizons would permit faster reactions to sudden physical shifts (e.g., Var. Env, Struct. Fail scenarios), our empirical results demonstrate that the representation stability provided by a richer temporal context outweighs raw responsiveness, ultimately yielding superior closed-loop adaptation.

*Table C.2.* Ablation of trajectory horizon length ($H$) across Walker2d and Ant environments. Rewards are averaged over 5 random seeds. Top-1 result is highlighted using bold numbers.

| Environment | $H$ | ID | OOD Damping | | Var. Env | Struct. Fail |
|---|---|---|---|---|---|---|
| | | | **0.3** | **2.2** | | |
| | 10 | 4217 | 4490 | 4512 | 4486 | 236 |
| | 30 | 4315 | 4510 | 4480 | 4530 | 393 |
| **Walker2d** | 50 (Ours) | 4884 | **5342** | **5349** | **5293** | 952 |
| | 70 | **5009** | 5121 | 5120 | 5137 | **1489** |
| | 10 | 4087 | 2021 | 1301 | 1688 | 868 |
| | 30 | 2263 | 2326 | 3012 | 1355 | **1733** |
| **Ant** | 50 (Ours) | **5184** | **4797** | **4789** | **3462** | 1003 |
| | 70 | 2961 | 3318 | 2440 | 2129 | 733 |

## C.2. Tuning Heuristics

We use reward scale to balance RL and representation objectives, and simply tune it such that the actor and decoder losses share similar magnitudes. We found this heuristic to work well across all environments.

For the encoder, we typically set $\beta = 0.05$ initially and reduce it if posterior collapse is observed. Across all environments, we find that the encoder KL loss stabilizes at approximately 1.3, which we suggest as a reference target when tuning $\beta$.

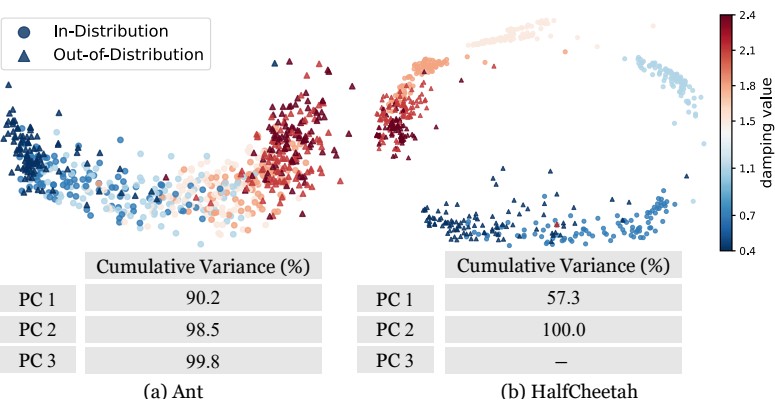

| | Cumulative Variance (%) | | | Cumulative Variance (%) |
|---|---|---|---|---|
| PC 1 | 90.2 | | PC 1 | 57.3 |
| PC 2 | 98.5 | | PC 2 | 100.0 |
| PC 3 | 99.8 | | PC 3 | − |
| | (a) Ant | | | (b) HalfCheetah |

*Figure D.1.* Latent topology via PCA. We plot explained variance of the first three PCs for the damping manifold. Ant exhibits an effectively 3D structure, while HalfCheetah collapses to 2D (PC3 negligible), consistent with their physical constraints.

# D. Additional Experiments

## D.1. Physical Isomorphism

Beyond structure discovery via functional equivalence, the learned latent space exhibits a topological complexity that mirrors the physical constraints of the agent. We analyze this geometric complexity by examining the effective dimensionality of the sub-manifold generated by varying a single parameter (damping). We apply Principal Component Analysis (PCA) to the latent codes to obtain cumulative variances. For Ant, we project the original latent code $z \in \mathbb{R}^5$ to $\mathbb{R}^3$ formed by the first 3 principal components, and for HalfCheetah $z \in \mathbb{R}^3$ we directly visualize the raw latent space. As visualized in Fig. D.1, the sub-manifold for HalfCheetah (a planar, 2D agent) collapses effectively to a 2D surface (cumulative variance reaches 100% within 2 components). In contrast, for Ant (an omnidirectional 3D agent), the damping sub-manifold spans a meaningful volume across 3 principal components within its 5-dimensional latent space. This suggests that LDG does not merely learn a generic 1D scale for parameters, but constructs a latent topology that is *isomorphic* to the agent's control complexity. For a simple 2D agent like HalfCheetah, the effect of damping can be adequately compressed into a 2D plane to represent amplitude and frequency shifts. However, for the 3D Ant, damping alters dynamics across multiple axes, affecting stability, yaw rotation, and leg coordination differentially. By maintaining a higher-dimensional (3D) representation for this single parameter, LDG avoids over-compressing these complex behavioral shifts into a simple scalar. This provides the policy with a behaviorally isomorphic context, allowing it to distinguish how the dynamics have shifted (e.g., differentiating between uniform drag vs. rotational instability), rather than just estimating that resistance has increased.

## D.2. Additional Baseline Comparisons

Context-Aware Dynamics Models (CaDM (Lee et al., 2020b)) serves as a direct baseline for our approach, as it also follows an outcome-centric paradigm and infers latent contexts from interaction histories. To rigorously evaluate the benefits of our explicitly geometric approach, we compare LDG against pure CaDM and a contrastively augmented variant (CaDM + InfoNCE), which applies our proposed contrastive objective to the CaDM representation.

As shown in Table D.1, pure LDG significantly outperforms pure CaDM across all environments and metrics. This validates the superiority of our geometry-aware approach over standard contextual forward-backward prediction. Interestingly, augmenting CaDM with our InfoNCE objective yields environment-dependent results. In Walker2d, it degrades performance, while in Ant environment, adding the contrastive loss substantially improves upon pure CaDM across all metrics, even surpassing pure LDG in the Structural Failure scenario. This demonstrates that our contrastive objective can successfully impart crucial topological structure to existing baseline representations, effectively enhancing their out-of-distribution robustness when objective gradients are properly aligned.

## D.3. Approximating Encoder Lipschitz Constant

To directly support Theorem A.1, we provide an empirical comparison of the Lipschitz constant between our method (LDG), a standard VAE, and Spectral Normalization (SN) in the Walker2d environment. The formal definition of $L_E$ (Definition A.6) relies on the Total Variation divergence of continuous window distributions, which is intractable to compute

*Table D.1.* Performance comparison of LDG against the CaDM baseline and a contrastively augmented variant (CaDM + InfoNCE). Rewards are averaged over 5 random seeds.

| Environment | Method | ID | OOD Damping | | Var. Env | Struct. Fail |
| | | | 0.3 | 2.2 | | |
|---|---|---|---|---|---|---|
| | CaDM | 3768 | 3990 | 4002 | 4005 | 225 |
| **Walker2d** | CaDM + InfoNCE | 3245 | 3726 | 3787 | 3313 | **1300** |
| | LDG (Ours) | **4892** | **5343** | **5336** | **5293** | 952 |
| | CaDM | 2467 | 1491 | 2319 | 1175 | 661 |
| **Ant** | CaDM + InfoNCE | 3790 | 2033 | 3006 | 2057 | **2049** |
| | LDG (Ours) | **5184** | **4797** | **4789** | **3462** | 1003 |

*Table D.2.* Empirical approximation of the encoder Lipschitz constant ($\hat{L}_E$) in the Walker2d environment.

| Method | $\hat{\mathbf{L}}_{\mathbf{E}}$ |
|---|---|
| LDG (Ours) | 7.84 |
| VAE | 20.77 |
| Spectral Normalization (SN) | 3.26 |

directly. Therefore, we estimate an empirical Lipschitz constant, $\hat{L}_E$, using local gradient perturbations over the sampled trajectory buffer. For a given batch of trajectory windows $w_i$, we approximate the spectral norm of the encoder's Jacobian using random, $L_2$-normalized projection vectors $\mathbf{v}$. The empirical Lipschitz constant is estimated by taking the supremum over the dataset and the projection vectors:

$$\hat{L}_E \approx \max_{w_i \in \mathcal{B}} \max_{||\mathbf{v}||_2 = 1} \left\| \nabla_{w_i} \left( \mathbf{v}^\top E(w_i) \right) \right\|_2 \tag{D.1}$$

where $\mathcal{B}$ represents the replay buffer, and $E(w_i)$ is the mean of the encoder's predicted latent distribution. It is important to note that this gradient-based metric is an imperfect approximation, discrepancy arises because our approximation measures local sensitivity on high-dimensional, empirical trajectory data distributions rather than computing exact global analytic bounds across the entire continuous space. Despite this inaccuracy, the *relative magnitudes* of $\hat{L}_E$ across different methods remain highly informative and serve as a reliable proxy for comparing geometric smoothness.

As shown in Table D.2, the purely reconstructive VAE exhibits a highly volatile latent space ($\hat{L}_E = 20.77$), making it highly susceptible to input perturbations. Our contrastive objective (LDG) significantly smooths this geometry ($\hat{L}_E = 7.84$), effectively bridging the gap to theoretical stability without explicitly restricting the network's weights. While Spectral Normalization yields the smoothest representation ($\hat{L}_E = 3.26$), as discussed in Section 5.4 in the main thesis, this rigid constraint prevents the formation of meaningful topological structure, ultimately degrading OOD adaptation.

