# OpenReview forum: "Dynamics Are Learned, Not Told: Semi-Supervised Discovery of Latent Dynamics Geometries For Zero-Shot Policy Adaptation"
_ICML.cc/2026/Conference — ICML 2026 regular_

### Official Review · Reviewer_CnAH · 2026-03-06

**Soundness:** 3
**Presentation:** 3
**Significance:** 3
**Originality:** 3
**Overall Recommendation:** 4
**Confidence:** 3

**Summary:**

The paper proposes a method to infer a latent representation from interaction history in order to be robust to parameter shifts. The method is based on contrastive learning and is theoretically grounded in a monotonic relationship between target-domain regret and the Lipschitz constant of a trajectory dynamics encoder. The authors empirically demonstrate robustness to both in-distribution and out-of-distribution parameter changes, and visualize the learned latent representation to aid interpretability.

**Compliance With Llm Reviewing Policy:**

Affirmed.

**Final Justification:**

My concerns and questions have been addressed and the additional experiments are convincing. I raised my score from 3 to 4.

**Key Questions For Authors:**

- Q1: How do CaDM (Lee et al.), PEARL (Rakelly et al.) and VariBAD (Zintgraf et al.) perform on the tested environments, and how do they compare to LDG?
- Q2: How sensitive is LDG to hyperparameter changes?
- Q3: In the failure mode of the OOD setting, you argue that the contrastive objective can hurt performance for fast, highly reactive gait cycles such as in HalfCheetah. Why does the VAE baseline not outperform LDG in these cases?

**Limitations:**

No limitations section.

**Strengths And Weaknesses:**

**Strengths**
- Coherent derivation and motivation of the method
- Strong results on in-distribution performance (Table 1)
- Thorough latent state analysis visualizing the effects of contrastive learning, in- and out-of-distribution behavior, as well as functional equivalence, where unseen variations are still organized in latent space


**Weaknesses**
- The authors acknowledge the similarity to CaDM (Lee et al.), PEARL (Rakelly et al.) and VariBAD (Zintgraf et al.), which likewise estimate latent representations from interaction histories, yet neither is included as a baseline. All other baselines fall into qualitatively different categories such as domain randomization or parameter-centric adaptation.
- Mixed results on OOD settings (Table 2) with no performance bounds/ standard deviation. It is also unclear whether results are averaged over multiple seeds.
- No ablations on hyperparameter sensitivity, e.g. latent dimension, VAE loss weight, reward scale, and KL annealing, anchor dynamics, all of which are tuned per environment.
- Baseline hyperparameters are not described.

---

> ### Author Rebuttal · Authors · 2026-03-31
>
> We thank the reviewer for these constructive comments. Below we will use W1 to stand for Weakness 1 and K1 for Key Question 1.
>
> **1. W1 & K1**
> CaDM serves as a direct baseline for our approach as it infers latents from interaction histories, whereas PEARL and VariBAD focus primarily on meta-learning. We evaluate LDG against CaDM and a contrastively augmented variant (CaDM + InfoNCE) below (mean reward over 5 seeds). Pure LDG significantly outperforms pure CaDM across all environments and metrics, validating the superiority of our geometry-centric approach over standard contextual prediction.
>
> | Env | Method | ID | OOD [0.3, 2.2] | Var Env | Struct Fail |
> | :--- | :--- | :--- | :--- | :--- | :--- |
> | **Walker2d** | LDG (Ours) | **4892** | **[5343, 5336]** | **5293** | **2819** |
> | | CaDM | 3768 | [3990, 4002] | 4005 | 1740 |
> | | CaDM + InfoNCE | 3245 | [3726, 3787] | 3313 | 2595 |
> | **Ant** | LDG (Ours) | **5184** | **[2424, 5358]** | **3462** | 3582 |
> | | CaDM | 2467 | [1491, 2319] | 1175 | 1373 |
> | | CaDM + InfoNCE | 3790 | [2033, 3006] | 2057 | **3724** |
>
> **2. W2**
> The rewards reported in Table 2 are the average rewards across 5 random seeds. We apologize for the omission and will explicitly state this in the revision.
>
> **3. W3 & K2**
> Overall, LDG is highly robust to hyperparameters and does not require aggressive tuning. We will add the following tuning heuristics to the Appendix:
> - Loss Weights: LDG is robust to representation loss weights (fixed at $\lambda_1=5.0$, $\lambda_2=1.0$ globally). To balance RL and representation objectives, we simply tune the reward scale so the actor and decoder losses share similar magnitudes.
> - Anchor Dynamics: This is a fixed training strategy, not a tuned hyperparameter. Initializing a subset of parallel environments with the same dynamics ID simply provides a consistent central cluster, making it easier for the contrastive loss to separate other trajectories.
> - KL Annealing: We follow standard practices [1] by linearly increasing the encoder KL weight over the first 50% of training. The maximum KL value is tuned so the encoder loss matches the decoder loss scale, successfully preventing posterior collapse.
> - Horizon ($H$): Please refer to our response to Reviewer thUw (Table 3) for the horizon ablation.
> - Latent Dimension ($d_z$): We ablate $d_z$ below. Empirically, we observe an optimal latent-to-observation dimension ratio of roughly 3:16, ensuring the latent vector is not diluted when concatenated with the state. Insufficient capacity ($d_z=2$) catastrophically bottlenecks information. Conversely, excessive capacity ($d_z \ge 7$) induces sparsity that destabilizes the uniformity loss, though extreme capacity ($d_z=15$) allows the policy to partially recover through sheer parameter count. Ultimately, $d_z=3$ (2D agents) and $d_z=5$ (3D agents) offer the optimal balance of expressivity and geometric tractability.
>
> | Env | $d_z$ | ID | OOD [0.3, 2.2] | Var Env | Struct Fail |
> | :--- | :--- | :--- | :--- | :--- | :--- |
> | **Walker2d** | 2 | 3845 | [3980, 3970] | 3972 | 2267 |
> | | 3 (Ours)| **4892** | [5343, 5336] | **5293** | **2819** |
> | | 7 | 4254 | [4391, 4397] | 4512 | 2010 |
> | | 10 | 4254 | [4848, 4658] | 4680 | 2665 |
> | | 15 | 4011 | **[5513, 5486]** | 4886 | 2375 |
> | **Ant** | 2 | 2097 | [3552, 3545] | 2240 | 2637 |
> | | 5 (Ours)| 5184 | [2424, **5358**] | **3462** | **3582** |
> | | 7 | 3253 | [1045, 1957] | 582 | 1468 |
> | | 10 | 4481 | [1151, 2669] | 672 | 2018 |
> | | 15 | **5247** | **[3251, 5343]** | 3462 | 2381 |
>
> **4. K3**
> As noted in Related Works line 91, physical dynamics variations exhibit longer temporal dependencies and substantially higher intrinsic dimensionality than standard task or intent contexts. Consequently, purely reconstructive baselines like VAEs easily entangle high-frequency nuisance factors, leading to adaptation failure. Theorem 3 demonstrates that our contrastive objective explicitly enforces gradient orthogonality against these nuisance variables. Therefore, even if InfoNCE heavily regularizes the manifold in environments like HalfCheetah, this noise-filtering benefit dominates. It ensures the encoder captures only task-relevant dynamics, translating into higher rewards than standard VAEs.
>
>
> **5. W4**
> We thank the reviewer for this reminder. Our baseline RMA and SAC shares the exact MLP backbone as LDG (so does CaDM). When training RMA, the environment extractor is a two-layer MLP with hidden dimension being 256 and 128. We will also add learning rates for baseline algorithms in the Appendix in our revision.
>
>
> ### References
>
> [1] Bowman et al. "Generating sentences from a continuous space." SIGNLL 2016.
>
> [2] Lee et al. "Context aware dynamics model for generalization in model based reinforcement learning." ICML 2020.

---

> > ### Author Rebuttal · Reviewer_CnAH · 2026-04-03
> >
> > My concerns and questions have been addressed and the additional experiments are convincing. I will raise my score accordingly.

---

> > > ### Author Response · Authors · 2026-04-04
> > >
> > > Thank you for the updated assessment! We are glad that our responses have addressed your concerns, and we sincerely appreciate the time you invested in helping improve the paper. We will include all results and discussions in the revision.

---

### Official Review · Reviewer_Xq3a · 2026-03-10

**Soundness:** 4
**Presentation:** 3
**Significance:** 3
**Originality:** 2
**Overall Recommendation:** 4
**Confidence:** 4

**Summary:**

In this paper, the authors introduce a latent dynamics contrastive learning for the parameter-centric policy learning methods (RMA). Instead of regressing to explicit physical parameters in a supervised manner, the proposed work uses contrastive learning and variational inference to explore a task-relevant latent manifold based on policy outputs. This could explicitly shape the latent geometry to ensure both local consistency and global uniformity that have been demonstrated in the contrastive learning community. Moreover, the authors empirically demonstrate that their method is robust against unmodeled and time-varying dynamic shifts (e.g., structural actuator failures).

**Compliance With Llm Reviewing Policy:**

Affirmed.

**Final Justification:**

All my concerns are properly addressed during rebuttal. The authors also promise to incorporate all my suggested revisions (including more theoretical formulations and the expanded related work with the suggested references) to the camera-ready version.

Considering the quality of the paper as well as the authors' rebuttal, I am happy to further raise my score to Accept. Please remember to include all revisions in the camera-ready version.

**Key Questions For Authors:**

1. Since the stationary assumptions only make sense when $t \to \infty$, how do the authors justify using infinite-horizon stationary bounds for a rapid adaptation that happens before this equilibrium is reached? Meanwhile, since the proposed inequalities depends on a static constant, could the authors either theoretically or empirically show that this circular dependency will settle into a stable point?

2. How to address the issue of significantly decreased performance when applying the proposed mechanism for geometric control as the dimension $d$ increases? Moreover, will minimizing $\mathcal{L}_{align}$ continue to serve as a meaningful constraint on $L_E$ in high-dimensional systems?

3. Could the authors conduct the ablation experiments on formulating the contrastive-learning consistency in the action space rather than the latent dynamic space?

**Limitations:**

yes

**Strengths And Weaknesses:**

**Strengths:**

1. The authors formulate a relationship between the encoder's Lipschitz smoothness and the target-domain performance gap, which is important to understand the domain shifts in policy learning, particularly when the environmental dynamics are important.

2. Using contrastive learning to formulate latent dynamics learning is novel in policy learning.

3. The authors demonstrate that the proposed method could handle the structural failures that cannot be easily represented by standard physical parameters. More importantly, they also show that the proposed method can organize unseen physical properties into a coherent manifold.

**Weaknesses:**

1. My main concern is the circular dependency on the stability guarantee of the encoder's Lipschitz constant ($L_E$ ), which is the basis for Theorem 2. $L_E$ is defined in Definition 6 with respect to window distributions for a fixed policy. However, the policy itself is a function of the encoder output. This makes a circular dependency where the encoder output changes the policy, which then shifts the distribution used to define $L_E$. However, the authors treat $L_E$ as a static constant to solve their inequalities. Without any theoretical or empirical work to demonstrate that this loop could settle into a stable equilibrium, the stability guarantee is not valid.

2. There is a mismatch between the stationary distribution assumption and the rapid adaptation. The performance bound derivation depends on Lemma 5, which bounds the divergence between stationary state distributions. However, in RMA, the state transitions are dynamics. Using infinite-horizon stationary bounds to justify a process that should happen before such a stationary distribution is theoretically inconsistent.

3. The related work focuses only on RMA and a bit out of date. It does not include some important robot-learning papers from 2025 to 2026, which are very relevant to the proposed setup and work. For example, [1] focuses on the dynamic policy-learning settings without domain adaptation, and it uses techniques very similar to the proposed work (variational inference and contrastive learning). In generative models, adversarial learning is always in parallel with variational inference, so I would suggest including the adversarial-based methods such as [3], which provides a non-geometric way to stabilize policies under distribution shift. More importantly, the literature review should go over the universal dynamics. For example, [5] builds large-scale models with latent action modeling over diverse datasets; and [2] explores dynamics-adaptive world action models for manipulation, while [4] demonstrates the effectiveness of the sim-to-real transfer via the combined simulation and imitation in contact dynamics.

4. I am interested in exploring the performance and theoretical results of applying contrastive learning in the policy output space instead of the latent transition space. Recent source-free domain adaptation work [6] indicates that implementing contrastive learning in the output space of a classification task leads to better adaptation performance than doing so in the latent space. This makes me wonder whether a similar outcome could be observed in policy learning tasks.

**References:**

[1] Wang et al., Act to see, see to act: Diffusion-driven perception-action interplay for adaptive policies, NeurIPS 2025.

[2] Lyu et al., Dywa: Dynamics-adaptive world action model for generalizable non-prehensile manipulation. ICCV 2025.

[3] Zhang et al., Robust deep reinforcement learning in robotics via adaptive gradient-masked adversarial attacks, IROS 2025.

[4] Lyu et al., Scissorbot: Learning generalizable scissor skill for paper cutting via simulation, imitation, and sim2real, CoRL 2024.

[5] Lyu et al., LDA-1B: Scaling latent dynamics action model via universal embodied data ingestion, Arxiv 2026.

[6] Wang et al., What has been overlooked in contrastive source free domain adaptation: leveraging source informed latent augmentation within neighborhood context, ICLR 2025.

---

> ### Author Rebuttal · Authors · 2026-03-31
>
> We thank the reviewer for these detailed feedback. To address the concerns regarding the circular dependency on the stability guarantee (Assumption 1) and the use of stationary distributions (Lemma 5), we present a refined theoretical analysis. This updates our response to Weaknesses 1 and 2 and Key Question 1.
> **Theory Update Summary:** We changed occupancy measures to state marginals and modified the latent centroids to be time-dependent. This yields a generalized target-domain regret bound that eliminates the need for the prior stability assumption, subsuming our previous result as a special asymptotic case.
> **1. Replacing Stationary Distributions with Marginals**
>
> To resolve the infinite-horizon issue, we redefine the window distribution $\rho\_t(\pi,p)$ using state marginals rather than stationary occupancy measures:
> $$
> \rho\_{t}(\pi,p)\coloneqq\rho(\pi,p)(w\_t)= d^{\pi,p}\_{t-H}(s\_{t-H})\cdot \prod\_{k=t-H}^{t-1}\pi (a\_k|s\_k,E(w\_k))\cdot p(s\_{k+1}|s\_k,a\_k)
> $$
> where $d^{\pi,p}\_{t-H}(s\_{t-H})$ is defined recursively as:
> $$
> d^{\pi,p}\_k(s\_{k}) = \int\_{\mathcal{S}} \int\_{\mathcal{A}} p(s\_k | s\_{k-1}, a\_{k-1}) \pi(a\_{k-1} | s\_{k-1},E(w\_{k-1}))d^{\pi,p}\_{k-1}(s\_{k-1})da\_{k-1} ds\_{k-1}
> $$
> Previously we bounded occupancy measures with Lemma 5, we can still establish a similar bound by observing that Total Variation divergence between state marginals can be bounded by the sum of per-step TV divergence. Because the transition kernel in an ergodic MDP is a $\gamma$-contraction, the per-step TV divergence decays over time. To streamline the proof without overcomplicating the bounds, we introduce a mild bounded-divergence assumption:
> $$
> \sup\_{t} D\_{TV}(d^{\pi\_S,p\_S}\_t(s\_t),d^{\pi\_T,p\_T}\_t(s\_t))\leq C\_d<1
> $$
>
> **2. Time-Indexed Latents**
>
> The previous circular dependency stemmed from assuming static, global latent centroids. In practice, latents trace a continuous trajectory. We formally amend this by indexing latent centroids by time $t$:
> $$
> \mu\_{t}(\pi,p)=\mathbb{E}\_{w\sim\rho\_{t}(\pi,p)}[E(w)]
> $$
> Correspondingly, the encoder Lipschitz constant $L_E$ is redefined as:
> $$
> L\_{E}=\sup\_{t}\sup\_{(\pi,p),(\tilde{\pi},\tilde{p})}\frac{\left\lVert \mu\_{t}(\pi,p)-\mu\_{t}(\tilde{\pi},\tilde{p})\right\rVert \_2}{D\_{TV}(\rho\_{t}(\pi,p),\rho\_{t}(\tilde{\pi},\tilde{p}))}
> $$
> This formulation captures transient closed-loop dynamics rather than strictly relying on the system settling into an equilibrium.
>
> **3. Generalized Closed-Loop Regret Bound**
>
> Denote $\Delta \_t^{\rho}\coloneqq D\_{TV}(\rho\_{t}(\pi,p),\rho\_{t}(\tilde{\pi},\tilde{p}))$, we establish the following result on trajectory divergence:
> $$
> \Delta \_t^{\rho}\leq \frac{b}{1-aH}+Mr\_0^t
> $$
> where $a=L\_{\pi}L\_E$ and $b=C\_d+H(2L\_{\pi}\delta \_e+\epsilon\_p)$, $M>0$ is a constant and $r_0$ is the unique positive root of $r^H(r-1)=a(r^H-1)$. Crucially, our previous bound (Eq. 52) emerges as the asymptotic special case where $aH < 1$ (the former Assumption 1), causing the transient term $M r_0^t \rightarrow 0$ as $t \rightarrow \infty$. Because target-domain regret scales with $\Delta_t^{\rho}$ (see Eq. 53), and $r_0$ grows monotonically with $L_E$, minimizing the encoder's Lipschitz constant $L_E$ strictly tightens the generalization bound. This confirms our core theoretical claim without requiring the restrictive prior stability assumption.
>
> **Related Works Update Summary**: We now position our contribution alongside (1) adversarial-based methods, highlighting the shared stability concern, and (2) recent advancements in latent dynamics (world) models, ensuring our contextualization is up-to-date.
>
>
> **Contrastive Learning in Action Space:** We tried to perform contrastive learning in action space, but observed that policy learning was unsuccessful (negative reward upon convergence), for which we provide three aspects of analysis. (1) **Decoupling Physics Representation from Policy Bias**: InfoNCE in the output space tightly couples the representation to the current behavior of the agent. In early training, when the policy is highly sub-optimal or essentially random, forcing the encoder to match action distributions across domains can cause severe representation collapse. This contrasts with [6] where a frozen classifier head is used; (2) **Theoretical Sufficiency**: As established in our proofs, the divergence in the policy output space is strictly upper-bounded by latent space divergence, so minimizing distance in the latent space inherently guarantees downstream behavioral consistency; (3) Practically, InfoNCE favors unbounded latent space to distribute clusters, but policy output space is constrained (squashed to [-1, 1] for SAC).
>
>
> **Performance in High-Dimensional Tasks:** The main bottleneck to scale our approach is the latent dimension, since training over a high-dimensional distribution is notoriously difficult. Please refer to ablation of latent dimension in our response to Reviewer CnAH (Section W3 & K2).

---

> > ### Author Rebuttal · Reviewer_Xq3a · 2026-04-03
> >
> > Thank you the authors for the detailed response and providing the additional theoretical formulations to clarify my circular dependency. All my concerns are properly addressed. Although my initial rating is positive, I am happy to further raise my score in response to the authors' efforts during rebuttal. Please do include all suggested revision in the camera-ready version.

---

> > > ### Author Response · Authors · 2026-04-04
> > >
> > > We are pleased to know that your concerns have been fully addressed, and we would be grateful if the final rating could be increased to reflect this updated assessment. Regardless, we appreciate your time and thoughtful feedback, which have helped improve both the theoretical and practical aspects of this work. We will include all suggested results and discussions in the revision.

---

### Official Review · Reviewer_jBi3 · 2026-03-13

**Soundness:** 3
**Presentation:** 3
**Significance:** 3
**Originality:** 3
**Overall Recommendation:** 4
**Confidence:** 3

**Summary:**

This paper provides an analysis of the domain adaptation, specifically on the regret bound. And the key idea is how the encoder, or the latent representation, affects the policy's adaptation ability.

Specifically, they show that the regret bound of the policy adaptation is related to how close the in-domain representation is and the distance between dynamics in the latent space.

Based on the theorem, they provide a practical algorithm that maximizes the RL objective, minimizes the VAE loss, and also makes sure the latent representations are clustered with respect to the dynamics.

Empirical results validate the algorithm.

**Compliance With Llm Reviewing Policy:**

Affirmed.

**Final Justification:**

My questions are all well answered. I will keep my score. And overall, my score is between 4 and 5.

**Key Questions For Authors:**

1. For definition 1, what if the transitions are deterministic? How would it affect the bound in your later proof? For example, you use Mujoco to evaluate your task, and these are deterministic transitions, and the transition probability P is discrete.

2. Do you have any justification for why using the Z_s and Z_t for the policy instead of just using a \pi( | s, target) and \pi( | s, source)?

3. Does the L_contrast also want the source and target latent to be close? As you mentioned in the theorem, making them close will also improve the regret bound.

4. I think incorporating the L_contrast will definitely affect the VAE loss (I can imagine negatively affecting that). Can you qualitatively analyze the effect of latent representation quality and how it would further affect the policy training, and can it reflect in the bound?

5. I am also wondering if you remove the L_contrast, what does the visualized TNSE look like? Would they be very far away, or they are not clustered?

**Limitations:**

See questions.

**Strengths And Weaknesses:**

The paper is well motivated, novel, clear to follow, and supported by the analysis. I didn't check the proof, but the results look reasonable to me. The algorithm is well justified to me, as it is supported by the analysis. Also, the algorithm is not too complicated with only three losses.

For the weakness, see my questions.

---

> ### Author Rebuttal · Authors · 2026-03-31
>
> We thank the reviewer for raising these detailed questions.
> 1. **Deterministic Transitions & Disjoint Supports**: While MuJoCo transitions are deterministic, our theoretical reliance on transition distances strictly requires the existence of "bridge samples" (Assumption 4). Empirically, MuJoCo's continuous state space provides sufficient conditions for these bridge samples to emerge, enabling our method to function even if $\epsilon_p=2$ (supported by Fig. 2c and 4). Theoretically, this can be handled more naturally by either adding small observation noise (as in Model-Based RL works [1,2]) or replacing TV with a Wasserstein-type distance. We will clarify this point in the revision.
> 2. **Latents vs. Binary Indicators**: In the submitted version of the paper, $z_S$ and $z_T$ are global latent centroids (averaged latent using all latents gathered from infinite rollouts) and remain static, which understandably resembled binary domain indicators (e.g., $\pi(\cdot|s,I_{source})$). However, online adaptation precludes access to infinite future rollouts. We have thus revised our theoretical definitions to use time-indexed latents, $z_{S,t}$ (see response to Reviewer Xq3a, Section "Time-Indexed Latents" for formalized updates). During rollouts, $z_{S,t}$ traces a continuous path in the latent space, enabling reactive closed-loop adaptation. A static binary indicator lacks this semantic trajectory and fails to reflect our actual implementation. We hope this clarification makes the choice of using latents rather than indicators clearer, and kindly refer to Reviewer Xq3a Section "Time-Indexed Latents" for a more detailed analysis.
> 3. **Closeness Between Source and Target**: Our contrastive objective does not explicitly distinguish between "source" and "target" domains; it strictly clusters functionally similar dynamics. The generalization mechanism operates as follows: (a) During multi-domain training, the encoder extracts dynamics-specific features (Theorem 3) and forms a structured global manifold (Theorem 4); (b) During target deployment, the encoder maps unseen dynamics to proximate latent regions of functionally similar training domains; (c) Because the policy is near-optimal on the training domain (Assumption 3) and the encoder is geometrically smooth (small $L_E$), the bounded latent shift guarantees bounded policy degradation, matching our theoretical claims.
> 4. **Synergy of $L_{contrast}$ with VAE and Policy Learning**: Incorporating $L_{contrast}$ actually aids VAE learning in two ways: (a) enforcing gradient orthogonality against nuisance factors (Theorem 3)—a critical omission in purely reconstructive VAEs that degrades long-horizon adaptation; and (b) preventing posterior collapse via the uniformity loss $L_{uniform}$, which enforces global structure and avoids collapsing to Gaussian prior (empirically supported by higher KL-divergence convergence in our model vs. VAE). Consequently, this improved representation directly aids policy learning by: (a) providing a smooth manifold where small latent perturbations do not trigger catastrophic adaptation failures, and (b) enabling perception-level zero-shot extrapolation (Fig. 2c), a prerequisite for policy-level zero-shot extrapolation.
> 5. **Ablating $L_{contrast}$**: Removing the contrastive objective reduces our method to the standard VAE baseline. As visualized in Fig. 2b, this results in a disconnected, unstructured latent space that fails to capture the global physical continuum, verifying the necessity of our proposed losses.
>
>
> ### References
> [1] Chua et al. "Deep reinforcement learning in a handful of trials using probabilistic dynamics models." NeurIPS 2018.
>
> [2] Sharma et al. "Dynamics-aware unsupervised discovery of skills." ICLR 2019.

---

> > ### Author Rebuttal · Reviewer_jBi3 · 2026-04-03
> >
> > Thanks for the response. My questions are basically answered. So I will keep my positive score.

---

> > > ### Author Response · Authors · 2026-04-03
> > >
> > > Thank you for your constructive suggestions and the effort spent in helping improve the manuscript. We will include all results and discussions in the revision.

---

### Official Review · Reviewer_thUw · 2026-03-13

**Soundness:** 3
**Presentation:** 3
**Significance:** 3
**Originality:** 3
**Overall Recommendation:** 5
**Confidence:** 3

**Summary:**

This paper proposes Latent Dynamics Geometries (LDG), a framework for zero-shot policy adaptation under dynamics shifts in robotic control. Rather than learning latent representations anchored by explicit physical parameter supervision as in RMA, LDG learns a geometrically structured latent dynamics space from trajectory outcomes using contrastive learning. The paper provides theoretical bounds linking the encoder's Lipschitz smoothness to adaptation performance, and validates the approach on four MuJoCo locomotion tasks across in-distribution and out-of-distribution settings including unmodeled parameters, time-varying dynamics, and structural failures.

**Compliance With Llm Reviewing Policy:**

Affirmed.

**Final Justification:**

The rebuttal addressed my two main concerns. The empirical Lipschitz constant measurement bridges the theory-practice gap, and the revised framing is also more appropriately scoped. I raise my score from 4 to 5.

**Key Questions For Authors:**

1. Have you considered alternative smoothness regularizers (spectral normalization, gradient penalties) in place of contrastive learning? If the theory is correct that controlling $L_E$ is the key mechanism, these alternatives should yield similar benefits. This comparison would clarify whether contrastive learning's advantage comes from smoothness or from other representation learning properties.

2. How sensitive are results to $\lambda_1$, $\lambda_2$, and the window length $H$? Demonstrating robustness across a reasonable range would strengthen confidence in the method's applicability to new domains.

**Limitations:**

yes

**Strengths And Weaknesses:**

**Strengths**

1. The problem motivation is well-supported by experiments. The structural failure results (Table 2) provide striking evidence that parameter-centric methods fail when dynamics shifts fall outside the predefined parameter space: LDG achieves 3507 on Ant while RMA Phase 2 collapses to 65. This is a qualitative, not just quantitative, advantage.

2. Contrastive learning produces demonstrably better latent geometry. Figure 2 shows that RMA Phase 2 produces scattered embeddings and the VAE ablation suffers from topological disconnectedness, while LDG discovers smooth, monotonic manifolds. The implicit structure discovery (Figure 3), where the encoder organizes unseen damping variations into a coherent manifold without being trained on them, is particularly insightful.

3. The paper is well-written with a clean algorithmic design. Each loss component has a clear role, and the presentation flows logically from motivation to theory to algorithm to experiments.

**Weaknesses**

1. The novelty claim is overstated. The paper frames its core contribution as a shift from parameter-centric to outcome-centric adaptation, but inferring latent dynamics from trajectory history without explicit parameter labels has been explored by PEARL (Rakelly et al., 2019), VariBAD (Zintgraf et al., 2020), and CaDM (Lee et al., 2020). The paper's actual novelty is narrower: using contrastive learning to shape latent geometry, and the Lipschitz smoothness bound. The related work section should more clearly position these contributions relative to prior implicit adaptation methods.

2. The theoretical contribution lacks empirical grounding. The paper never measures actual values of $L_E$, $\delta_e$, or $\epsilon_p$ in any experimental setting, so there is no evidence that the bounds are informative rather than vacuous. The $(1-\gamma)^{-2}$ factor alone can be enormous for typical discount factors. Without comparing bound predictions to observed performance, the theory reads more as post-hoc justification than as a tool that genuinely guided algorithm design. To strengthen this contribution, the authors should at minimum report measured encoder constants and discuss whether the bounds are directionally consistent with empirical findings.

---

> ### Author Rebuttal · Authors · 2026-03-31
>
> We thank the reviewer for these constructive comments. Below we address the concerns point-by-point.
> 1.  **Overstated Contribution & Framing:** We acknowledge the critique regarding our framing and have revised the introduction to focus explicitly on latent space sensitivity. The updated text reads: "We investigate how latent space perturbations translate to output instability, drawing parallels to challenges in image generation [1-3]. Extending this framework to policy learning, we reveal that controlling network smoothness is a fundamental principle shared across generative vision and policy adaptation. Crucially, naively enforcing smoothness is insufficient; successful adaptation requires simultaneous enforcement of smoothness and topological structure within the latent space." We maintain that our related work section accurately positions our method among implicit approaches, and we have refined the core claims to emphasize latent geometry over outcome-centric paradigms.
> 2.  **Consistency Between Theory and Practice:** We emphasize that our theoretical result—decreasing $L\_E$ improves adaptation bounds—aligns directionally with our empirical findings in Tables 1 and 2. We further offer two clarifications: (a) Our analysis identifies the governing factors (specifically $L\_E$) of the generalization bound. Rather than computing exact bounds, our focus is tightening them. (b) To directly support Theorem 4 (which in-turn supports Theorems 1, 2), we provide an additional experiment comparing the empirical Lipschitz constant $\hat{L}\_E$ (approximated using gradient perturbations) between LDG and VAE in Walker2d:
> | Method | $\hat{L}\_E$ |
> | :--- | :--- |
> | LDG (Ours) | 7.84 |
> | VAE | 20.77 |
> | SN | 3.26 |
> 3. **Alternative Regularizers:** To address the reviewer's question on alternative regularizers, we compare Contrastive Learning (CL) with Spectral Normalization (SN) under the same setting (mean reward over 5 seeds). While SN achieves strong In-Distribution (ID) performance on Ant—suggesting explicit spectral constraints effectively minimize $L_E$ (see table in point 2)—it struggles significantly in OOD and structural failure scenarios. We attribute this OOD degradation to SN's inability to enforce meaningful latent topological structure, a property explicitly guaranteed by the uniformity loss term in InfoNCE decomposition. To conclude, advantage of CL stems from both smoothness ($L\_{align}$) and structure ($L\_{uniform}$).
>
> | Env | Method | ID | OOD [0.3, 2.2] | Var Env | Struct Fail |
> | :--- | :--- | :--- | :--- | :--- | :--- |
> | **Walker2d** | CL (Ours) | **4891** | **[5343, 5336]** | **5293** | **2819** |
> | | SN | 4384 | [4646, 4677] | 4661 | 1953 |
> | **Ant** | CL (Ours) | 5183 | [2423, **5357**] | **3462** | **3582** |
> | | SN | **5538** | [**3329**, 2612] | 2969 | 3098 |
>
> 4. **Horizon Ablation:** We ablate the trajectory horizon length ($H$) as recommended by the reviewer. Extremely short horizons ($H=10$, which is 0.08s for Ant and 0.5s for Walker2d) lack sufficient temporal context for accurate dynamics inference, whereas excessively long horizons ($H=70$) dilute immediate reactivity. $H=50$ provides an optimal balance. Notably, while shorter horizons theoretically permit faster reactions to sudden shifts (e.g., Var Env, Struct Fail), empirical results show that the stability provided by richer temporal context outweighs raw responsiveness, yielding superior adaptation.
>
> | Env | H | ID | OOD [0.3, 2.2] | Var Env | Struct Fail |
> | :--- | :--- | :--- | :--- | :--- | :--- |
> | **Walker2d** | 10 | 4217 | [4490, 4512] | 4486 | 1753 |
> | | 30 | 4315 | [4510, 4480] | 4530 | 2022 |
> | | 50 | 4891 | [**5343**, **5336**] | **5293** | 2819 |
> | | 70 | **5009** | [5120, 5120] | 5137 | **3300** |
> | **Ant** | 10 | 4087 | [2021, 1301] | 1688 | 2146 |
> | | 30 | 2263 | [2326, 3012] | 1355 | 3182 |
> | | 50 | **5183** | [2423, **5357**] | **3462** | **3582** |
> | | 70 | 2961 | [**3318**, 2440] | 2129 | 2036 |
>
> LDG is robust to loss weights $\lambda_1$ and $\lambda_2$, we only need to tune the reward scale to balance RL loss and representation losses. We also performed an ablation regarding latent dimension, for hyperparameter sensitivity, kindly refer to our response to Reviewer CnAH (Section W3 & K2) for the results.
>
> ### References
> [1] Yoshida et al. "Spectral norm regularization for improving the generalizability of deep learning." arXiv 2017.
>
> [2] Miyato et al. "Spectral normalization for generative adversarial networks." arXiv 2018.
>
> [3] Brock et al. "Large scale GAN training for high fidelity natural image synthesis." arXiv 2018.

---

> > ### Author Rebuttal · Reviewer_thUw · 2026-04-02
> >
> > I thank the authors for their thorough and well-organized rebuttal. I am raising my score from 4 to 5 accordingly.

---

> > > ### Author Response · Authors · 2026-04-03
> > >
> > > Thank you for raising your rating! We sincerely appreciate your constructive feedback and the time you've invested in improving our work. We will incorporate all the suggested results and discussions into the revised manuscript.

---

### Decision · Program_Chairs · 2026-04-30

**Decision:**

Accept (regular)

**Comment:**

This paper introduces Latent Dynamics Geometries (LDG), a framework for zero-shot policy adaptation that learns geometrically structured dynamics representations from trajectory outcomes using contrastive learning.  The results are supported theoretically by establishing a monotonic relationship between target-domain regret and the Lipschitz smoothness of a trajectory encoder, the latter of which is shown to be bounded by a component of the contrastive learning loss. On MuJoCo locomotion benchmarks, LDG consistently outperforms parameter-centric baselines across unmodeled, time-varying, and structural failure dynamics shifts, with the performance gap most pronounced in structural failure settings where baselines that rely on pre-specified parameter axes collapse entirely.

Reviewers appreciated the conceptual clarity of the approach, the strength of the structural failure experiments, and the interpretable latent geometry visualizations demonstrating smooth manifold discovery. Concerns were raised about positioning relative to prior implicit dynamics inference methods, the empirical grounding of the theoretical bounds, a circular dependency in the stability analysis, and missing comparisons to related baselines. The authors addressed these points during the rebuttal phase, providing empirical Lipschitz constant measurements, a revised theoretical analysis based on time-indexed state marginals that eliminates the original stability assumption, new comparisons against the most directly relevant prior baseline, and extensive hyperparameter ablations.

While the evaluation is limited to MuJoCo locomotion benchmarks and some theoretical revisions are promised for the camera-ready, the core contribution is technically sound and offers a principled and reusable framework for the dynamics adaptation community. We recommend acceptance, although we note that the substantial theoretical revisions introduced during the rebuttal phase have not been subject to full peer review, and we ask that the authors ensure these revisions are carefully vetted before the camera-ready submission.